# Cryo-EM structure of cell-free synthesized human histamine 2 receptor/G$_s$ complex in nanodisc environment

Zoe Köck[1,6], Kilian Schnelle[2,3,6], Margherita Persechino[4], Simon Umbach[1], Hannes Schihada[4], Dovile Januliene[2,3], Kristian Parey[2,3], Steffen Pockes[5], Peter Kolb[4], Volker Dötsch[1], Arne Möller[2,3] ✉, Daniel Hilger[4] ✉ & Frank Bernhard[1] ✉

Here we describe the cryo-electron microscopy structure of the human histamine 2 receptor (H$_2$R) in an active conformation with bound histamine and in complex with G$_s$ heterotrimeric protein at an overall resolution of 3.4 Å. The complex was generated by cotranslational insertion of the receptor into preformed nanodisc membranes using cell-free synthesis in *E. coli* lysates. Structural comparison with the inactive conformation of H$_2$R and the inactive and G$_q$-coupled active state of H$_1$R together with structure-guided functional experiments reveal molecular insights into the specificity of ligand binding and G protein coupling for this receptor family. We demonstrate lipid-modulated folding of cell-free synthesized H$_2$R, its agonist-dependent internalization and its interaction with endogenously synthesized H$_1$R and H$_2$R in HEK293 cells by applying a recently developed nanotransfer technique.

The biogenic amine histamine is a hormone and neurotransmitter that is ubiquitously distributed in the human body. It plays central roles in diverse (patho)physiological processes, such as inflammation, allergy, gastric acid secretion, cellular migration, vasodilatation, bronchoconstriction and neurotransmission[1,2]. Histamine signaling is mediated through binding and activation of the histamine 1–4 receptor subtypes H$_1$R, H$_2$R, H$_3$R and H$_4$R belonging to the family of G protein-coupled receptors (GPCRs)[1]. The H$_2$R is an important regulator of gastric acid secretion, ionotropic and chronotropic cardiac stimulation, vasodilatation and mucus production[1,3]. Histamine binding to the H$_2$R increases both cAMP and inositol phosphate second messengers by signaling through the heterotrimeric G proteins G$_s$ and G$_q$, respectively[4–6]. Clinical drugs targeting H$_2$R such as famotidine, nizatidine and cimetidine are applied to suppress gastric acid secretion in esophageal reflux disease and are important for peptic, gastric and duodenal ulcer healing[1,5,7]. More recently, H$_2$R is discussed as therapeutic target for

several cardiovascular conditions as well as for acute myeloid leukemia, diabetes and colorectal cancer[8,9]. The proposed H$_2$R homodimerization and agonist-dependent cross-desensitization, co-internalization and heterodimerization with the H$_1$R may provide additional routes for new therapeutic strategies[10]. A central challenge, however, has been the development of subtype-selective ligands with low off-target side effects. Thus, a more detailed structural basis of ligand binding specificity and H$_2$R activation is required to design drugs that can precisely tune H$_2$R signaling.

Cell-free (CF) expression enables the production of proteins in the presence of stabilizing ligands and it allows the direct insertion of nascent membrane proteins into defined membrane environments. The CF synthesis of GPCRs was continuously improved during the last decade[11–13]. In particular, the strategy to insert nascent GPCRs cotranslationally into nanodiscs (NDs) avoids any contact with potentially denaturing detergents[11,14–17]. The insertion into empty ND membranes

[1]Centre for Biomolecular Magnetic Resonance, Institute for Biophysical Chemistry, Goethe-University of Frankfurt/Main, Frankfurt, Germany. [2]Department of Biology/Chemistry, Structural Biology section, University of Osnabrück, Osnabrück, Germany. [3]Center of Cellular Nanoanalytic Osnabrück (CellNanOs), University of Osnabrück, Osnabrück, Germany. [4]Department of Pharmaceutical Chemistry, University of Marburg, Marburg, Germany. [5]Institute of Pharmacy, University of Regensburg, Regensburg, Germany. [6]These authors contributed equally: Zoe Köck, Kilian Schnelle. ✉e-mail: arne.moeller@uni-osnabrueck.de; daniel.hilger@pharmazie.uni-marburg.de; fbern@bpc.uni-frankfurt.de

is translocon independent and is accompanied by a release of lipids[18]. The lipid composition of ND membranes can be of importance for the insertion efficiency as well as for the function of the inserted membrane proteins[13,16,19]. The additional presence of ligands as well as of interacting G proteins can further stabilize CF synthesized GPCRs and support their functional folding. By using these technical advantages, the formation of stable GPCR/G protein complexes in CF reactions was recently demonstrated[13].

Here, we use CF expression of the human $H_2R$ with *E. coli* lysates to determine the cryo-electron microscopy (cryo-EM) structure of the receptor in its active conformation and in complex with the $G_s$ heterotrimer at a global resolution of 3.4 Å. The complex is formed with full-length $H_2R$, $G_s$ and the $G_s$ stabilizing nanobody Nb35 in ND membranes composed of DOPG lipid. The biochemical characterization is complemented by analysis of the CF synthesized $H_2R$ after transfer into membranes of HEK293T cells using a recently developed nanotransfer technique[20–22].

## Results

### CF expression optimization and $H_2R/G_s/Nb35/ND$ complex preparation

Full-length human $H_2R$ was cotranslationally inserted into supplied preformed NDs by CF expression. Lipid type and charge can influence the membrane insertion efficiency and subsequent folding of a nascent membrane protein[13,23]. Therefore, initial expression screens were performed with a $H_2R$-mNG derivative and a set of NDs assembled with different lipids, and the resulting mNG fluorescence in the supernatant was analyzed as a measure for the overall $H_2R$-mNG solubilization. The screen included PC lipids, common in eukaryotes, and negatively charged PG lipids due to their potential to enhance the efficiency of translocon-independent membrane integration[24]. A similar effect has been described for cardiolipin, which increases membrane fluidity and may therefore also affect membrane insertion of the $H_2R$[25]. In addition, the effect of cholesterol was analyzed, as it is able to stabilize some GPCRs. Cardiolipin and cholesterol do not form ordered bilayers and were thus analyzed as additive in DOPG membranes. All ND types were supplied at final concentrations of 60 μM and reaction concentrations of solubilized $H_2R$-mNG between 5 μM and 35 μM were obtained (Supplementary Fig. 1a). The negatively charged lipids DOPG or DEPG were identified as being most efficient for $H_2R$-mNG insertion (Supplementary Fig. 1a). Besides quantity, the quality of the synthesized GPCR is of crucial importance and size-exclusion chromatography (SEC) can be used to separate functionally folded GPCR/ND particles from soluble but aggregated fractions[13]. SEC analysis was performed with CF synthesized $H_2R$ without the mNG moiety and purified $H_2R$/ND complexes showed the best sample quality with ND membranes composed of DOPG, DEPG and DMPG (Supplementary Fig. 1b). SEC profiling was then used to monitor additional effects of various supplied ligands on the quality of cotranslationally synthesized $H_2R$/ND (DOPG) complexes (Supplementary Fig. 1c). Stabilizing effects of the proposed folded $H_2R$/ND fraction were observed after synthesis in presence of the agonist histamine and with most antagonists (Supplementary Fig. 1c).

Cryo-EM samples of $H_2R$ in the active state and complexed to the heterotrimeric G protein $G_s$ were then produced by CF synthesis of $H_2R$ in presence of the agonist histamine, heterotrimeric $G_s$ protein purified from insect cells, preformed NDs (DOPG) and Nb35 containing a C-terminal His-tag (Nb35-His) in a total reaction volume of 1.8 mL (Fig. 1). After incubation for approx. 16 h, the reaction was treated with apyrase and the synthesized $H_2R/G_s/Nb35$-His complexes in NDs (DOPG) were purified by IMAC and subsequently analyzed by SDS-PAGE (Supplementary Fig. 2a, b) and SEC analysis (Supplementary Fig. 2c). The peak SEC fraction was taken for cryo-EM analysis and concentrated to 2.8 mg/mL in a final volume of 35 μL.

### Nanotransfer and oligomerization of CF synthesized $H_2R$ in HEK293

The functionality of CF synthesized $H_2R$ was analyzed using a recently developed technique for GPCR transfer from ND membranes into membranes of living cells[20,21] (Fig. 2a). Ligand binding was demonstrated by the histamine dependent internalization of transferred $H_2R$-mNG derivatives in HEK293 cells via an increase in the cytosolic fluorescence (Fig. 2b–d). Moreover, some co-localization of internalized transferred $H_2R$-mNG with $H_1R$-mCherry synthesized after transfection of the HEK293 cells indicate that receptors from different origins follow the same internalization route.

$H_2R$ is known to form stable homo-oligomers, in addition to hetero-oligomerization with the related receptor $H_1R$[10,26,27]. To study these interactions, CF synthesized GPCRs containing a C-terminal Strep-tag were transferred into HEK293 cells that were previously transfected with expression vectors encoding for Flag-tagged GPCR derivatives (Fig. 2e). Strep-tagged $H_2R$, Strep-tagged $H_1R$ or Strep-tagged free fatty acid 2 receptor (FFAR$_2$) serving as negative control were inserted into NDs (DOPG) by CF expression, purified and transferred into HEK293 cells previously transfected with constructs encoding for Flag-$H_1R$ or Flag-$H_2R$. After washing and cell lysis, the transferred GPCRs were immobilized by anti-Strep antibodies and co-immobilized GPCRs were identified by anti-Flag antibodies (Fig. 2e). The results revealed both, homodimerization of $H_1R$ and $H_2R$ as well as the $H_1R$-$H_2R$ heterodimerization, while no interactions of transferred Strep-tagged FFAR$_2$ with any of the histamine receptors was detected (Fig. 2e). These data agree with previous reports on the oligomerization of the tested GPCRs and further indicate the correct folding of the CF synthesized GPCRs after their transfer in HEK293 cells.

### Structure determination of the histamine/$H_2R/G_s$ signaling complex

We determined a cryo-EM structure of the histamine-bound $H_2R/G_s$ complex stabilized by Nb35 and embedded in ND (DOPG) membranes at a global resolution of 3.4 Å (Fig. 3 and Supplementary Fig. 3a, b). Representative class averages and the data analysis flow chart are shown in Supplementary Fig. 4, and associated statistics are summarized in Supplementary Table 1. The cryo-EM map shows an evenly distributed resolution with strong density for the bound agonist histamine and for the TM domains of $H_2R$ (Supplementary Fig. 5). Therefore, the cryo-EM map enabled building of an atomic model for the histamine-bound active conformation of $H_2R$ (residues D13$^{1.27}$-G298$^{8.53}$; superscript numbers indicate generic GPCR numbering following the revised Ballesteros-Weinstein system for family A GPCRs) including the canonical seven transmembrane helices (TMs 1–7) connected by three extracellular loops (ECLs 1–3) and three intracellular loops (ICL1–3), a common architecture of GPCRs (Supplementary Fig. 6). Due to the conformational heterogeneity of the C-terminal end of helix 8 (H8) and the resulting weak density, the receptor was only modeled until residue G298$^{8.53}$. The increased structural flexibility in this region could be a consequence of the missing palmitoylation of residue C304$^{8.59}$ at the C-terminal end of H8 of the CF-expressed receptor that is predicted to anchor H8 to the lipid membrane[28–30]. Otherwise, the local resolution map shows a relatively evenly distributed quality across the entire receptor complex with weaker density for the central part of ECL2, which remains partially unresolved (residues T164$^{ECL2}$-T171$^{ECL2}$) most likely due to its conformational flexibility. The C-terminal region of ECL2, however, is stabilized by one conserved disulfide bond connecting residue C91$^{3.25}$ in the N-terminal end of TM3 with residue C174$^{45.50}$ in ECL2. Additionally, we performed an alignment of our experimental structure with a computationally predicted model of the human $H_2R$ in complex with Gα$_s$. The model was generated by the AlphaFold2 (AF2)[31] multimer tool via the COSMIC2 platform[32] using the sequences for the human $H_2R$ and guanine nucleotide-binding protein G(s) subunit alpha

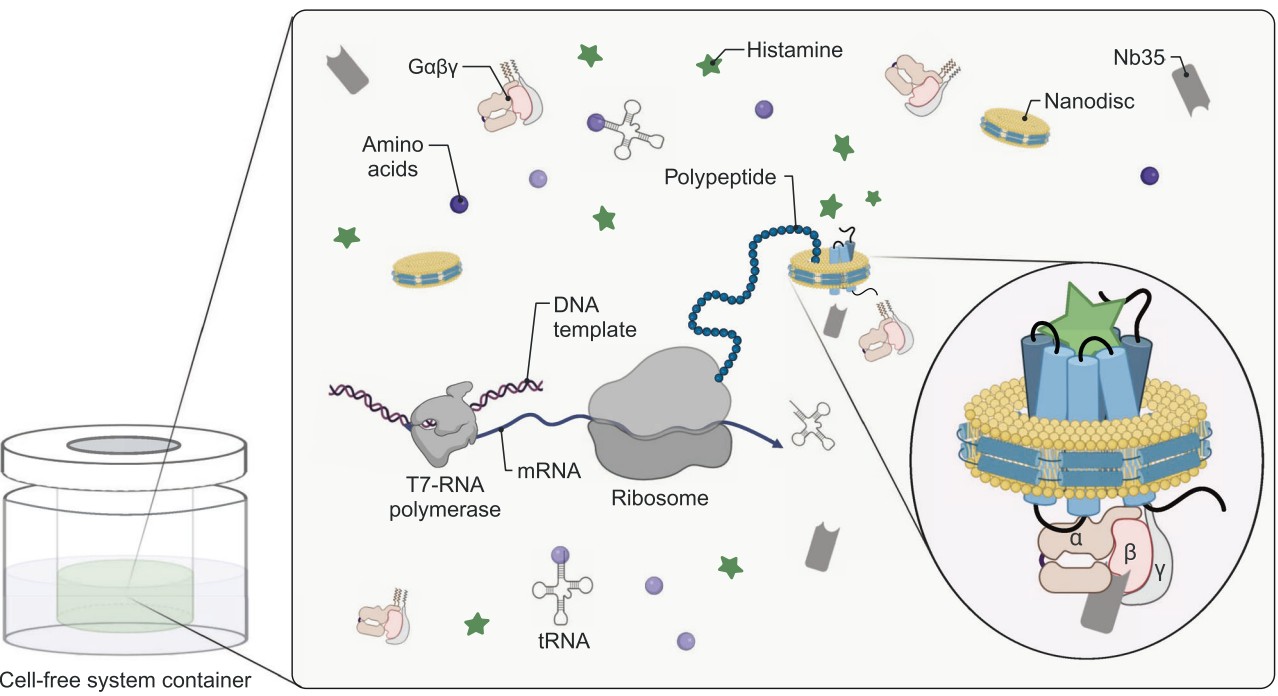

**Fig. 1 | Cryo-EM sample preparation of complexes by CF expression.** $H_2R$ is synthesized by cotranslational insertion into preformed NDs (DOPG) in the presence of histamine and $G_s$ heterotrimer purified from insect cells (Created with BioRender.com).

as reported in the UniProt database. The overall superposition of backbone atoms and side chains of the experimental and predicted complexes shows a rather high degree of congruence, with a Cα-root-mean-square deviation (RMSD) of the receptor portion of 1.046 Å. (Supplementary Fig. 3c).

## Comparison of histamine binding between the $H_2R$ and $H_1R$

The bound histamine shows clear density within the orthosteric ligand binding pocket of $H_2R$ formed by transmembrane domains (TMs) 3, 5, 6 and 7 (Fig. 4, Supplementary Fig. 5). The primary amine of histamine (Nα) forms H-bonds and ionic interactions with the backbone carbonyl and the carboxyl group of D98[3.32] in TM3, respectively—a highly conserved residue among aminergic receptors. Another H-bond interaction is formed between the Nπ atom of the imidazole ring and the conserved Y250[6.51] in TM6 (Fig. 4a, Supplementary Fig. 6). The rest of the histamine binding pocket is composed of hydrophobic aliphatic or aromatic residues in TM3 (V99[3.33]) and TM6 (Y250[6.51], F251[6.52], F254[6.55], and W247[6.48]) as well as polar side chains in TM3 (T103[3.37]), TM5 (D186[5.42], T190[5.46]) and TM7 (Y278[7.43]).

The affinity of histamine to $H_2R$ (p$K_i$ 6.6) is approx. 10-fold higher if compared with the affinity to $H_1R$ (p$K_i$ 5.6)[33,34]. Comparison of the $H_2R$-$G_s$ structure with the available active structure of the $H_1R$-$G_q$ complex (PDB ID 7DFL)[35] revealed some receptor subtype specific characteristics (Fig. 4a). As a common feature, the nitrogen atoms Nα and Nπ of the bound histamine engage in polar interactions with residues D[3.32] and Y[6.51], respectively. However, amino acid sequence variations in the orthosteric ligand binding site between the two receptor subtypes cause a rotation of the histamine imidazole ring in the $H_2R$ by approximately 80° with respect to the ligand binding pose in the $H_1R$. The rotation prevents a clash of the side chain of Y[3.33] in $H_1R$ (V[3.33] in $H_2R$) with the perpendicularly oriented imidazole ring in the $H_2R$ structure and causes an overall deeper binding position towards the $H_1R$ receptor core compared to the $H_2R$. As a result, the Nπ atom of histamine is able to form H-bonds with T112[3.37] in TM3 and N198[5.46] in TM5 of $H_1R$. In contrast, the distances between the Nπ atom of the histamine and the corresponding residues T103[3.37] in TM3 and T190[5.46] in TM5 of $H_2R$ are too large (4.9 Å and 6.5 Å, respectively) to allow hydrogen bonding

interactions. Furthermore, the non-conserved residue D186[5.42] in TM5 of the $H_2R$ is not in H-bonding distance to the ligand and therefore does not directly contribute to histamine-receptor interactions. This agrees with previous functional studies showing that this residue is crucial for $H_2R$-specific antagonist, but not histamine binding[36].

Based on these observations, subtype-specific residues in the $H_2R$ and $H_1R$ were tested by mutagenesis for their contribution to histamine-dependent receptor-mediated G protein activation. Residues at positions 3.33, 4.57, 5.42 and 5.46 in $H_2R$ were individually mutated to the corresponding residues in $H_1R$ and vice versa, and functionally characterized using bioluminescence resonance energy transfer (BRET)-based G protein activation (G-CASE) sensors[37] (Fig. 4b). All $H_1R$ mutations maintained > 40% of wild-type (WT) surface expression levels, while the expression levels of the $H_2R$ mutations S150W[4.57], D186T[5.42], and T190N[5.46] were significantly reduced or almost completely abolished (Supplementary Fig. 7). Therefore, transfected DNA of the WT $H_2R$ was adjusted accordingly to allow comparison at similar expression levels. The $H_2R$ mutations V99Y[3.33], S150W[4.57], and T190N[5.46] completely ablated $H_2R$-mediated $G_s$ activation. Consistent with the $H_2R$-$G_s$ complex structure and the functional studies mentioned above[36], D186T[5.42] had no impact on histamine-dependent receptor signaling. While low receptor amount in the plasma membrane might contribute to the missing activity of the T190N[5.46] and S150W[4.57] mutants, the effect of the V99Y[3.33] and S150W[4.57] mutations are presumably due to the larger size of the introduced tyrosine and tryptophan side chains. In particular, the substitution of V99[3.33] by tyrosine might result in steric hindrance of histamine binding in the $H_2R$, while a bulky tryptophan side chain at position 4.57 in TM4 presumably causes distortion of the ligand binding pocket due to clashes with residues in TM3. In the $H_1R$, mutations Y108V[3.33], W158S[4.57], and T194D[5.42] completely ablated receptor-mediated $G_q$ activation, whereas the substitution N198T[5.46] led to an 8-fold lower histamine potency (Fig. 4c). Besides the conserved common polar contacts between histamine and residues D[3.32] and Y[6.51] in all histamine receptors, these results confirm that the $H_1R$ and $H_2R$ exhibit subtype-specific receptor-ligand interactions that may also contribute to the molecular basis of agonist binding selectivity.

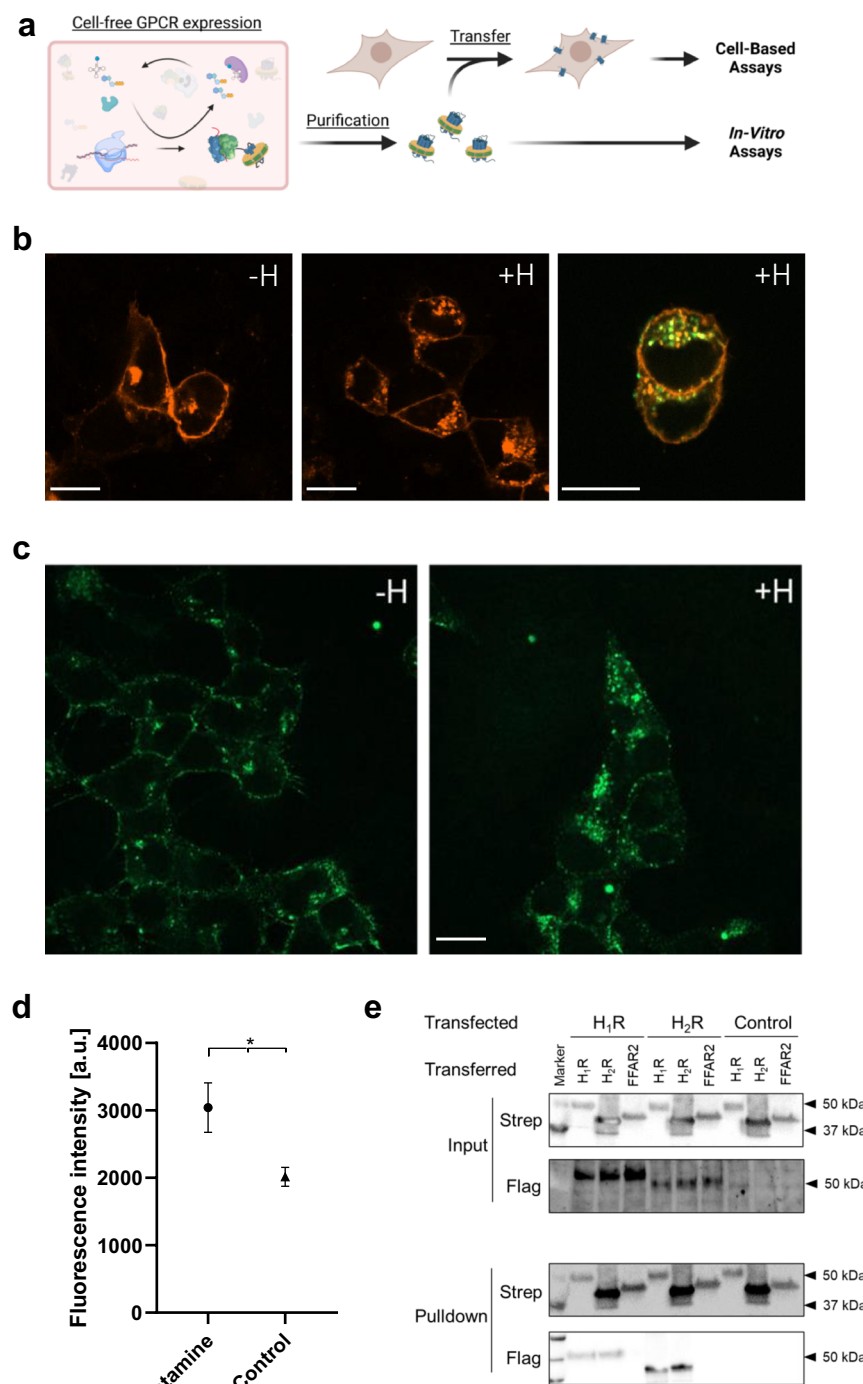

**Fig. 2 | Characterization of CF synthesized H₂R after nanotransfer into HEK293 cells. a** Schematic of GPCR nanotransfer. **b** H₁R-mCherry synthesized after transfection internalizes upon histamine treatment (100 μM histamine for 1 h; +H) and co-localizes with nanotransferred and CF synthesized H₂R-mNG (right panel). White bar = 10 μm. Representative images from three independent experiments. **c** HEK293 cells transferred for 4 h with 0.5 μM purified H₂R-mNG in NDs (DOPG) and treated with histamine. Left panel: Localization of transferred H₂R-mNG in untreated cells. Right panel: Internalization of transferred H₂R after incubation with 100 μM histamine for 1 h. White bar = 10 μm. Representative images from three independent experiments. **d** Quantification of cytosolic fluorescence of transferred H₂R-mNG. Data are presented as mean (SD) ($n$ = 3 individual experiments with ≥ six

data points each, *$P < 0.05$, two-sided students t-test). Source data are provided as Source Data file. **e** Interaction of transferred CF synthesized GPCRs with transfected H₁R and H₂R. Transfected cells expressing Flag-tagged H₁R or H₂R were transferred with CF synthesized Strep-tagged H₁R, H₂R, or FFAR₂ for 4 h. Subsequently, cells were washed, lysed and transferred GPCRs were immobilized by anti-Strep pulldowns. Input: Immunoblots of cell lysates before anti-Strep pulldowns. Pulldown: Immunoblots of immobilized GPCRs after anti-Strep pulldowns. Strep: Anti-Strep antibody; Flag: Anti-Flag antibody; Control: Cells without transfection. The experiment was performed once, uncropped figures are provided in the Source Data File. (Fig. 2a, created with BioRender.com).

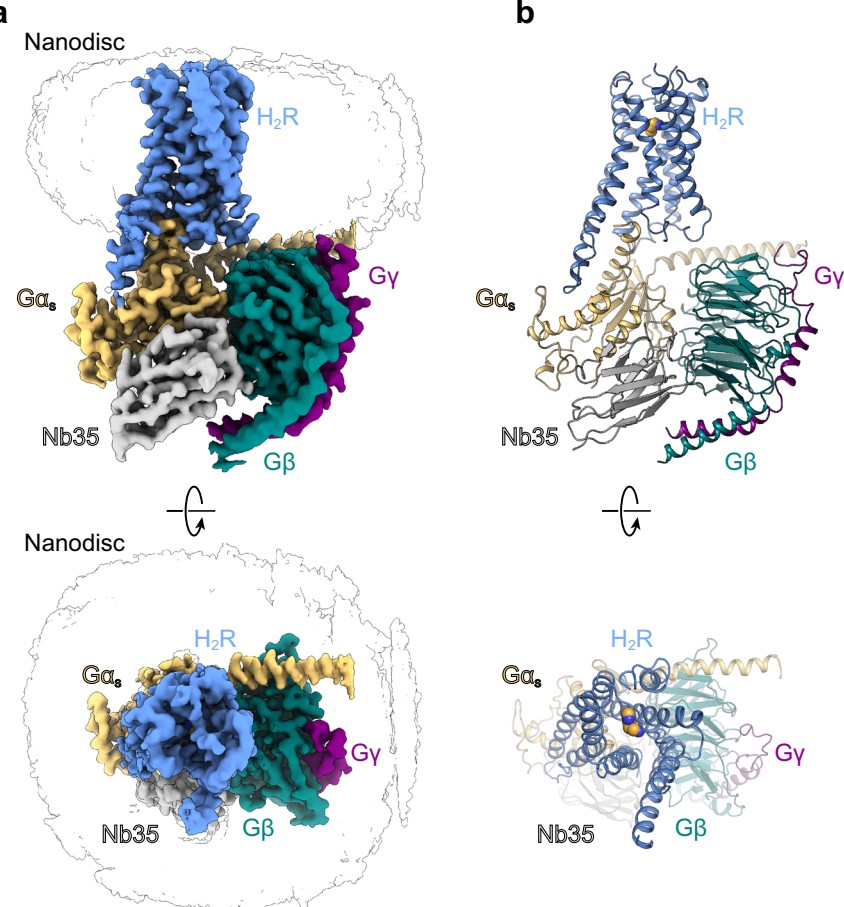

**a**
Nanodisc
H₂R
Gαs
Gγ
Nb35
Gβ
Nanodisc
Gαs
H₂R
Gγ
Nb35
Gβ

**b**
H₂R
Gαs
Gγ
Nb35
Gβ
H₂R
Gαs
Gγ
Nb35
Gβ

**Fig. 3 | Cryo-EM structure of the histamine-bound H₂R-Gₛ/Nb35/ND complex.**
**a** Cryo-EM density map of the histamine-bound H₂R/Gₛ/Nb35/ND complex colored by subunit. Blue, H₂R; wheat, Gαₛ; cyan, Gβ; purple, Gγ, grey, Nb35; black silhouettes, ND. **b** Ribbon model of the H₂R/Gₛ/Nb35/ND complex bound to histamine (yellow spheres) colored by subunit.

## H₂R activation by histamine

Compared to the inactive structure of the H₂R bound to the antagonist famotidine (PDB ID 7UL3)[38], the orthosteric ligand binding pocket in histamine-bound H₂R undergoes some subtle structural rearrangements caused by differences in receptor-ligand interactions (Fig. 5). Specifically, the H-bond between the imidazole nitrogen Nπ of the agonist and $Y^{6.51}$ in TM6, which is absent in the famotidine-bound inactive structure, seems to cause an inward movement of the extracellular end of TM6 (1.8 Å as measured between the Cα atoms of G258). Furthermore, the 3.8 Å downwards shift of the primary amine group of the agonist towards the intracellular side in comparison to the corresponding amine of the antagonist enables the formation of stronger H-bond interactions of the agonist with the backbone and side chain of the conserved $D98^{3.32}$ in TM3. This interaction might result in the observed counter-clockwise rotation of TM3 upon receptor activation. Another activation-dependent conformational change on the extracellular side of the receptor involves the slight movement of TM5 towards TM4. Notably, while histamine does not engage TM5, the guanidinium group of famotidine forms H-bond interactions with residues $D186^{5.42}$ and $T190^{5.46}$ in TM5 as well as $T103^{3.37}$ in TM3, which presumably prevents movement of these transmembrane helices and stabilizes the receptor in the inactive state.

More pronounced conformational changes take place on the intracellular side typical for the activation of family A GPCRs (Fig. 5)[39]. This includes the outward rotation of the cytoplasmic end of TM6 and concerted movements of the intracellular end of TM5 towards TM6 as well as the inward shift of the intracellular part of the adjacent TM7, resulting in the creation of the intracellular G protein-binding site (Fig. 5). Previous studies have identified conserved sequence motifs in family A GPCRs, also known as microswitches that are important for transmitting the ligand-induced conformational changes from the orthosteric ligand binding pocket to the G protein-binding cavity[39,40]. Comparison of structural alterations in those microswitch regions between the inactive and active structures of the H₂R, the H₁R and the β₂ adrenergic receptor (β₂AR) revealed nearly identical rearrangements, suggesting that these aminergic receptors share a similar activation mechanism (Supplementary Fig. 8). In particular, the 'toggle-switch' residue $W^{6.48}$ in the $C^{6.47}W^{6.48}xP^{6.50}$ motif at the bottom of the orthosteric ligand binding pocket is displaced upon receptor activation. Nearer to the intracellular side of the receptor, the $P^{5.50}I^{3.40}F^{6.44}$ motif undergoes a ratchet like motion that includes an outward movement of $F^{6.44}$ past residue $I^{3.40}$. Altogether, these conformational changes eventually induce the outward rotation of TM6 to enable opening of the intracellular G protein-coupling cavity and subsequent engagement of the heterotrimeric G protein. The formation of the active state is further facilitated by rearrangements of the highly conserved $D^{3.49}R^{3.50}Y^{3.51}$ motif and the $N^{7.49}P^{7.50}xxY^{7.53}$ motif on the intracellular side of TM3 and TM7, respectively.

## Agonist binding selectivity of H₂R and H₁R

H₁R and H₂R appear to have a conserved histamine recognition motif involving main contacts to residues $D^{3.32}$ and $Y^{6.51}$, whereas variation in the interaction with TM5 at position 5.42 or 5.46 may have a major contribution to agonist selectivity. To investigate the molecular basis

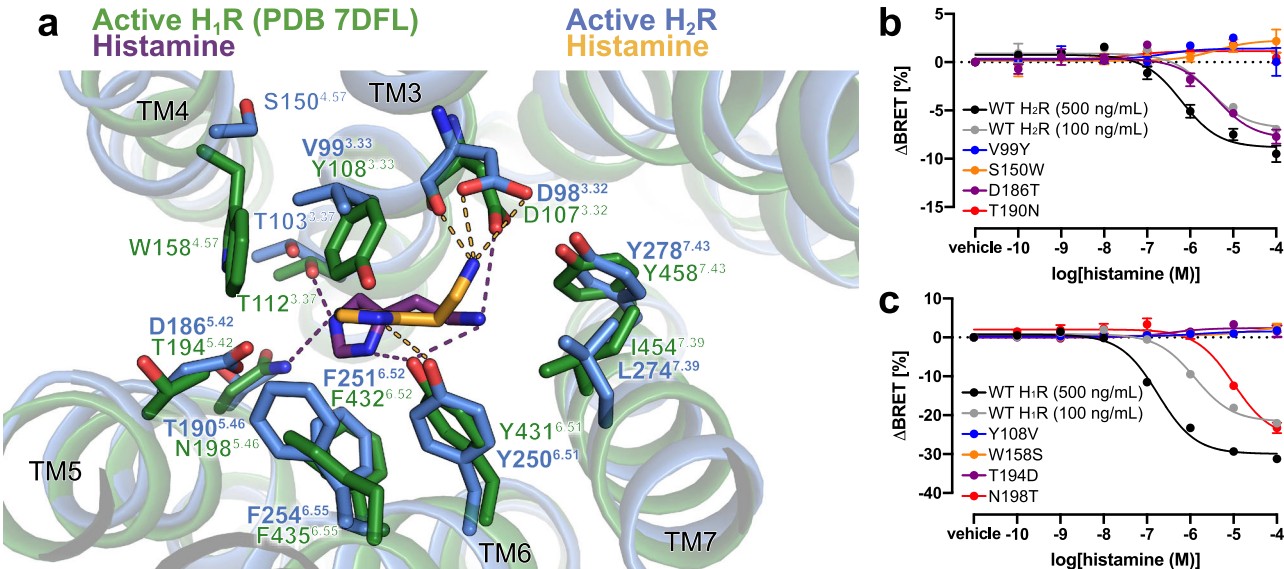

**Fig. 4 | Histamine-binding to H₂R. a** Comparison of the histamine-binding site in H₂R (blue, receptor; yellow, histamine) and H₁R (green, receptor; purple, histamine) (PDB ID 7DFL). Binding pocket residues are represented as sticks and labeled with residue number and Ballesteros-Weinstein code (superscript) colored by the receptor subtype. H-bond interactions are shown as dotted lines colored by the interacting ligand. **b, c** Mutagenesis analysis of receptor-specific residues within the histamine-binding pocket in H₂R **b** and H₁R **c** using a BRET-based G protein dissociation assay. Values in brackets represent the amount of DNA (ng) of receptor plasmid used to transfect one mL of cells. 500 ng DNA per mL of cells was used for all mutants. Signaling graphs show mean ± s.e.m. of three independent biological replicates with a global fit of the data. Source data are provided as Source Data file.

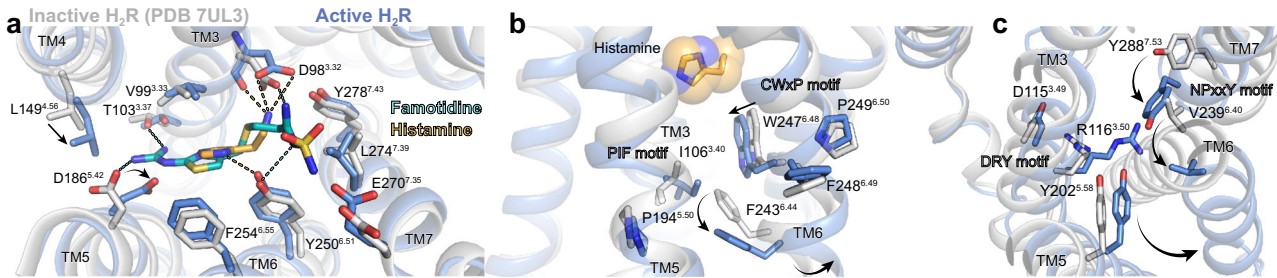

**Fig. 5 | H₂R activation by histamine. a** Binding pocket of the histamine-activated H₂R (blue) with the inactive famotidine-bound receptor (gray) (PDB ID 7UL3) overlaid. H-bond interactions are shown as dotted lines colored by the interacting ligand. **b** Close-up view of the CWxP and PIF motif in the inactive (grey) H₂R and the active (blue) H₂R bound to histamine (yellow spheres). **c** Comparison of the DRY motif and the NPxxY motif in the inactive (grey) H₂R and active (blue) H₂R structures. Amino acid side chains are represented as sticks and labeled with residue number and Ballesteros-Weinstein code (superscript) colored by the receptor subtype. Arrows indicate conformational changes in the H₂R upon activation.

of selective binding of H₂R-specific agonists, we first analyzed predicted binding modes by generating a variety of possible histamine orientations in the H₂R using SEED (Supplementary Fig. 9a). Considering the predicted binding energy of each pose, the canonical histamine orientation was indeed the top-scored one. Comparison of further predicted histamine binding modes in both receptors with the respective experimental ones showed that the docking software AutoDock Vina generated best-matching poses with a RMSD of the heavy atom positions of 0.76 Å for the H₁R and 0.57 Å for the H₂R (Supplementary Fig. 9b).

AutoDock Vina was then used to predict the binding modes for various H₂R-selective agonists in both receptors. The agonist amthamine engages in charge-assisted H-bond interactions with the conserved residue D98³·³² as well as with the H₂R-specific residue D186⁵·⁴² (Fig. 6a, b). In contrast, in the H₁R, amthamine is predicted in a different orientation and interacts with D107³·³² via the uncharged amino group, an interaction that lacks the ionic character, and has no direct contact with T194⁵·⁴² in TM5 (D186⁵·⁴² in the H₂R). In addition, the methyl group of amthamine is predicted to engage in a favorable

hydrophobic interaction with the side chain of F251⁶·⁵² in the H₂R that is not observed in the H₁R. Moreover, the presence of the bulky residue Y108³·³³ in TM3 of the H₁R seems to prevent any of the investigated H₂R-selective agonists from adopting a binding mode that is similar to the respective ones found in the H₂R (Fig. 6b).

To experimentally validate the predicted complexes, we used H₁R and H₂R mutants that were designed by swapping the receptor-specific amino acids in TM3-TM5, as described above, and performed BRET-based signaling assays in comparison to the WT receptors in the presence of amthamine (Fig. 6c, d). Notably, the H₂R mutant T190N⁵·⁴⁶ was excluded from these experiments because of its lack of expression and inactivity (Supplementary Fig. 7). Similar to histamine (Fig. 4b, c), the H₂R mutations V99Y³·³³ and S150W⁴·⁵⁷ completely abolished amthamine-stimulated Gs activation and supported the proposed steric hindrance of agonist binding. However, signaling of the mutant H₂R-D186T⁵·⁴² was remarkably different from the one stimulated by histamine, as it was completely unresponsive to amthamine (Fig. 6c). This abolished activity is most likely caused by disrupting the interaction between D186⁵·⁴² and the aromatic amine on the amthamine

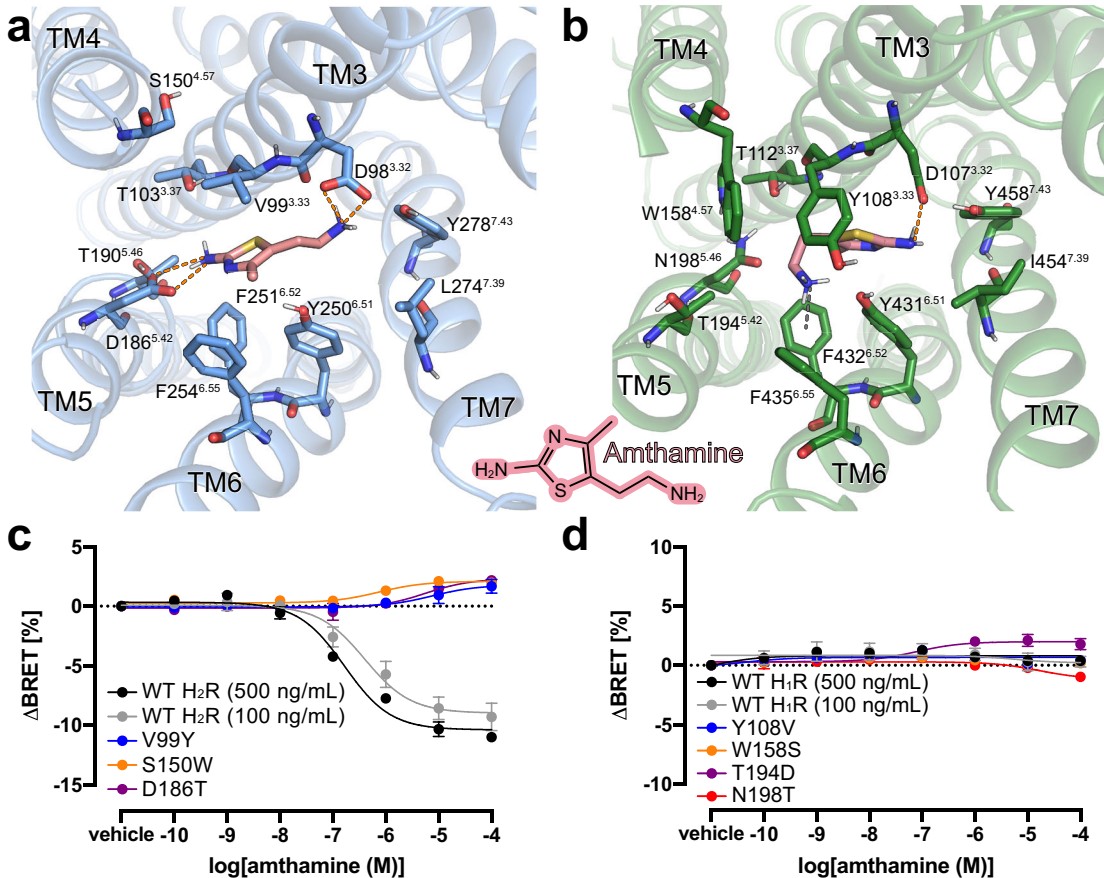

**Fig. 6 | Agonist binding selectivity of H₂R and H₁R. a** and **b** Predicted binding modes for the H₂R selective agonist amthamine (light pink) in **a** H₂R (blue) and **b** H₁R (green). The orange dotted lines represent H-bond interactions. The grey dotted line represents a possible cation-aromatic interaction between F432^6.52 in H₁R and the protonated amino group of amthamine. Binding pocket residues are represented as sticks and labeled with residue number and Ballesteros-Weinstein code (superscript). Binding mode predictions were obtained with AutoDock Vina.

**c**, **d** Mutagenesis analysis of receptor-specific residues within the ligand-binding pocket in **c** H₂R and **d** H₁R using a BRET-based G protein dissociation assay in the presence of amthamine. Values in brackets represent the amount of DNA (ng) of receptor plasmid used to transfect one mL of cells. 500 ng DNA per mL of cells was used for all mutants. Signaling graphs show mean ± s.e.m. of three independent biological replicates with a global fit of the data.

thiazole ring. For the H₁R WT and mutants, we observed no or only marginally amthamine-dependent stimulation of G_q signaling (Fig. 6d).

In addition to amthamine, possible binding modes in the H₁R and H₂R were also predicted for the H₂R-selective agonists dimaprit and the carbamoylguanidine derivative, compound 157[41] (reported K_i selectivity H₂R/H₁R ratio of 1:3802) (Fig. 7 and Supplementary Tables 2 and 3). As in the case with amthamine, the interaction with D186^5.42 can be identified as the main predicted reason for H₂R selectivity of both compounds. In summary, these functional studies in combination with the predicted binding modes of H₂R selective agonists underline the importance of positions 5.42 and 5.46 not only for selective binding of H₂R blockers as shown previously[36], but also for the investigated H₂R agonists.

## G protein coupling to the H₂R

The overall structure of the histamine-activated H₂R-G_s complex in a lipid ND shows a similar receptor-G protein conformation compared to other available aminergic receptor-G_s complex structures obtained in detergent (RMSD of 0.9 to 2.7 Å) (Supplementary Fig. 10). While all aminergic receptor-G_s complexes display the prototypical receptor-G protein conformation, they also exhibit significant differences in the relative orientation of the coupled G protein with respect to the receptor. Specifically, the N-terminal αN helix of the Gα_s subunit is rotated up to approximately 35° in the membrane plane relative to the

TM bundle between different receptor-G protein complexes. In the H₂R-G_s complex, the relative orientation of the αN helix is most similar to the one found in the complex structures of the adrenergic receptor β₂AR (PDB IDs 3SN6, 7BZ2, 7DHI, 7DHR), and the dopamine receptor D₁R (PDB ID 7F1Z) (Supplementary Fig. 10b). Five positively charged residues, K8, K17, K24, R13 and R20, in the αN helix of the H₂R-coupled Gα_s subunit are found to point towards the ND lipid bilayer to potentially interact with the polar membrane headgroups (Supplementary Fig. 10c). Notably, previous studies have shown that basic residues in the αN helix of Gα_s play important roles for the plasma membrane localization of the G protein and β₂AR-mediated activation of G_s in negatively charged lipid environment[42,43]. Furthermore, similar electrostatic interactions between basic residues in the αN helix and the lipid bilayer have been found in G_i complex structures of the dopamine receptor and the neurotensin receptor 1 NTSR1[44] in NDs, demonstrating that these N-terminal polybasic regions might play an important role for membrane anchoring and receptor-mediated activation of different G protein families in agreement with previous mutagenesis and functional studies[42,45].

The interface between the H₂R and G_s includes a buried surface area of 2951 Å². Like in other aminergic receptor-G_s complexes, the main interaction sites occur between ICL2 and TM3, TM5, TM6 and the TM7-H8 kink on the receptor and the αN-β1 hinge loop and α5 on the Gα subunit of the G protein. Upon coupling to the H₂R, the α5 helix of

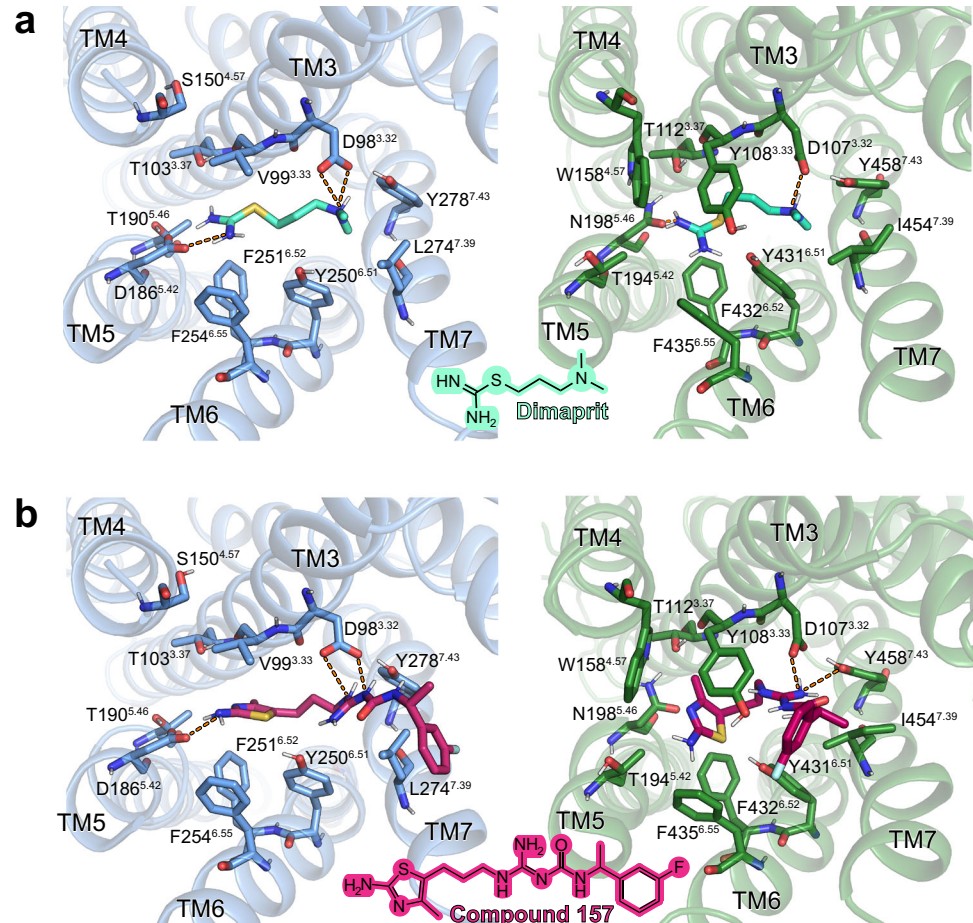

**Fig. 7 | Molecular docking of H₂R specific agonists in the H₂R and the H₁R.**
**a** Docking of dimaprit; **b** Docking of compound 157[41]. Structural elements and relevant residues of the binding pockets of the H₂R (blue) or the H₁R (green) are shown. The orange dotted lines represent H-bond interactions. Binding pocket residues are represented as sticks and labeled with residue number and Ballesteros-Weinstein number (superscript). Binding mode predictions were obtained with AutoDock Vina.

the G protein rotates by 60° and straightens up due to a 13 Å translation of its distal C-terminal end towards TM6 in comparison to the GDP-bound inactive structure of $G_s$[46] (Supplementary Fig. 10d). This movement is accompanied by a 5 Å shift of α5 towards the receptor to form a number of hydrophobic and hydrophilic contacts with residues in the intracellular G protein binding cavity of the H₂R. Together with the interaction between ICL2 of the receptor and the αN-β1 hinge loop and α5 helix of the G protein, which has been shown to impact the conformational dynamics of the β1 strand and the adjacent nucleotide-binding P loop of the $G\alpha_s$ subunit in complex with the β₂AR[47], the rotational translation of the α5 helix leads to a disruption of the GDP-binding pocket and subsequent nucleotide release.

## Comparison of the H₂R-G$_s$ and the H₁R-G$_q$ G protein-coupling interface

When the structures of the H₂R-G$_s$ and the H₁R-G$_q$ complexes are superimposed on the receptor, the $G\alpha_s$ and $G\alpha_q$ proteins display differences in their orientation relative to the TM bundle (Fig. 8a). The Ras domain of $G\alpha_s$ is shifted and slightly rotated away from TM3 towards TM6 when compared to the one of $G\alpha_q$. This difference is propagated to the αN helix and the Gβγ subunits of the G protein, resulting in significant differences in the receptor-G protein interactions relative to the H₁R-G$_q$ complex. In particular, the α5 helix is more shifted towards the TM5/TM6 region of the H₂R than in the H₁R-G$_q$ complex. The closer positioning of α5 of $G\alpha_s$ relative to TM5 and TM6 presumably leads to the slightly further outward displacement of TM6 in the H₂R compared

to the H₁R (Fig. 8a). Furthermore, the tighter interaction between α5 and TMs 5 and 6 allows the formation of a H-bond interaction between the backbone carbonyl of E392 of $G\alpha_s$ and the side chain of T235⁶·³⁶ as well as potential polar interactions of D381, Q384, and R385 of the G protein with residues in TM5 (Q212⁵·⁶⁸ and R215⁵·⁷¹) (Fig. 8b). Together with the van der Waals contacts formed between residues Y358 and I216⁵·⁷² as well as L346 and I219^ICL3 of $G\alpha_s$ and the receptor, respectively, these interactions putatively stabilize the more extended α-helical structure of the C-terminal end of TM5 in comparison to the TM5 of H₁R, which forms weaker contacts in this region with the engaged $G\alpha_q$ (Fig. 8c).

In addition to the H-bond between E392 of $G\alpha_s$ and T235⁶·³⁶ of the H₂R, already mentioned above, the C-terminus of the G protein α5 forms additional interactions with the TM5/TM6 region as well as the TM7/H8 loop and H8 of the H₂R (Fig. 8b). While the very C-terminal residue L394 in the "hook" of the $G\alpha_s$ α5 helix could not be modeled presumably due to its high flexibility, the adjacent conserved L393 is buried in a hydrophobic pocket lined by residues I205⁵·⁶¹, T235⁶·³⁶, L236⁶·³⁷ in TMs 5 and 6 of the H₂R and L368 of the α5 helix. Furthermore, residues R389, Q390, and E392 of the G protein are engaged in H-bond interactions with two arginine residues, R293⁸·⁴⁸ and R296⁸·⁵¹, located in the TM7/H8 hinge region and in H8 of the H₂R, respectively. This interaction potentially causes the shift of the backbone of R293⁸·⁴⁸ away from the receptor and closer to α5 in comparison to the H₁R-G$\alpha_q$ complex to engage in H-bonding with the G$\alpha_s$. For G$\alpha_q$, similar interactions are being formed between L358 and hydrophobic residues in

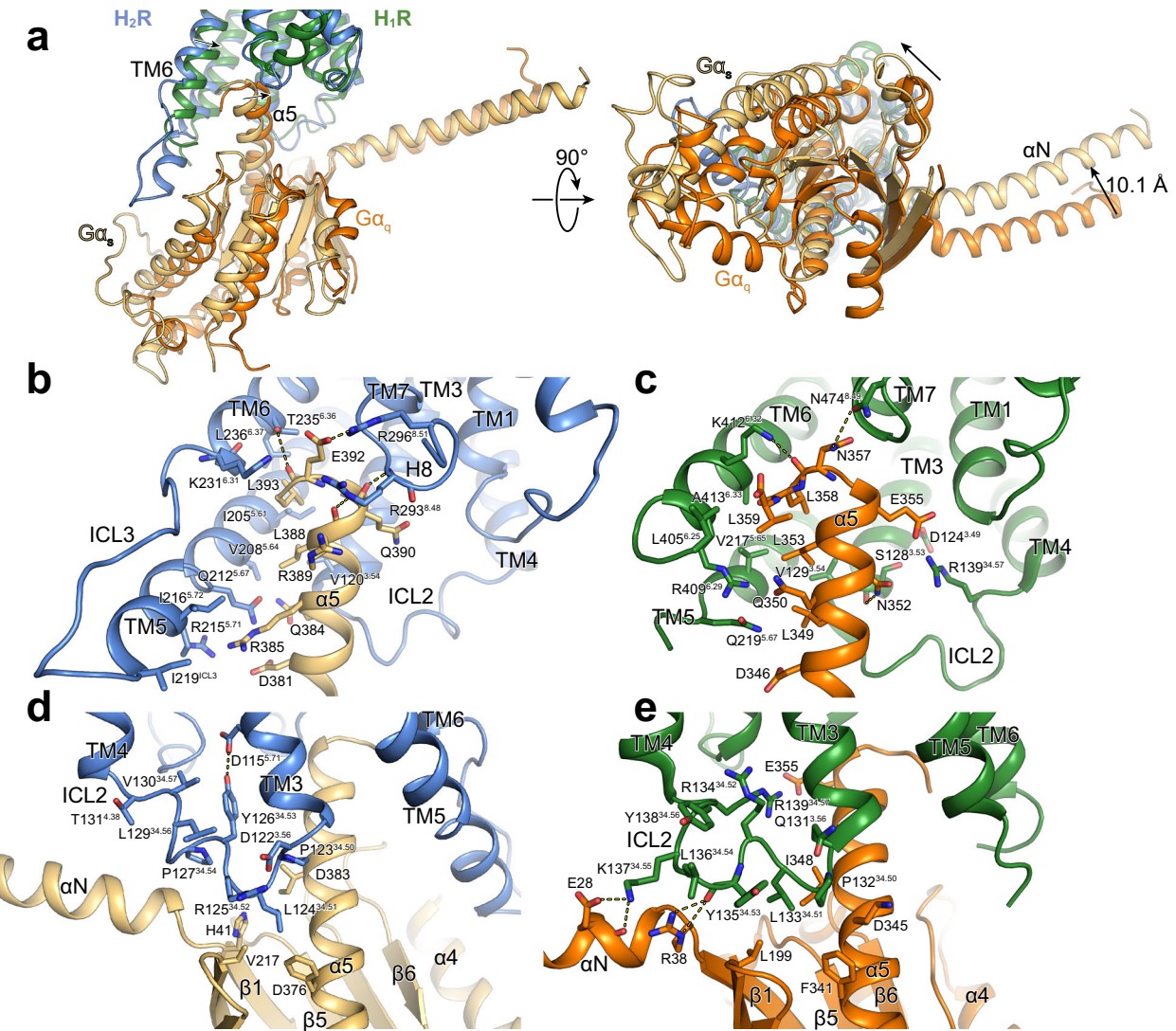

**Fig. 8 | Comparison of G protein interfaces between the H₂R-G_s and the H₁R-G_q complex. a** Comparison of the relative receptor-G protein orientation between the H₂R (blue) and the H₁R (green) subtype coupled to Gα_s (wheat) and Gα_q (orange), respectively. **b, c** Comparison of the interface between **b** the α5 of Gα_s and H₂R and **c** the α5 of Gα_q and H₁R. **d, e** Comparison of the interface between the ICL2 and the G protein subtype in **d** the H₂R-Gα_s complex and **e** the H₁R-Gα_q complex. Interface residues are represented as sticks and labeled with residue number and for the receptor residues with the Ballesteros-Weinstein code (superscript). Arrows indicate differences in the relative orientation of the G protein subtypes with respect to the receptors.

TMs 5 and 6 of the H₁R as well as N357 and N474[7.31] in the TM7/H8 hinge region of the receptor. Overall, however, the α5 of Gα_q seems to engage in fewer interactions with TM5 and the TM7/H8 region of H₁R compared to G_s with H₂R (Fig. 8c).

Additional differences between the two complex structures are also observed in the ICL2-G protein interactions. Specifically, in ICL2 of the H₂R-G_s structure, the conserved L124[34.51] is positioned deeply inside the hydrophobic pocket formed by residues F376 and I383 in α5 as well as V203 in β3 and H41 in β1 of Gα_s (Fig. 8d). As seen in other aminergic receptor-G_s complex structures, the ICL2 of H₂R forms a helix that is stabilized by an interaction between the conserved residues Y126[34.53] in the middle of the loop and D115[3.49] of the conserved DRY motif in TM3. In contrast, ICL2 of the H₁R adopts a position farther away from the Gα subunit so that L133[35.51] does not reach as deep into the hydrophobic pocket between α5, β1 and β3 of Gα_q as in the H₂R-G_s complex (Fig. 8e). Instead, the ICL2 of H₁R forms H-bonds between K137[34.55] and E28 and R134[34.52] and D124[3.49] of the DRY motif. The homologous residue to Y126[34.53] in the H₂R (Y135[34.53] in the H₁R) does not interact with the DRY motif in the H₁R-Gα_q structures but forms an H-bond interaction with

its backbone carbonyl and R38 in the αN-β1 loop region of the G protein. In summary, these differences in the receptor-G protein interface might explain some of the underlying molecular principles of the G protein-coupling specificity observed for the H₁R and the H₂R histamine receptor subtypes[48–53].

## Discussion

Here, we report a CF production pipeline to obtain a cryo-EM structure of a GPCR/G-protein complex. A CF lysate from *E. coli* strain A19 previously defined by proteomics analysis[54] was used in combination with supplied preformed lipid NDs, heterotrimeric G_s protein and Nb35 to obtain a structure of the full-length H₂R-G_s complex with its endogenous agonist histamine. The determined nominal resolution of 3.4 Å is comparable to analogous structures of GPCR complexes expressed and purified from eukaryotic cell-based systems[44,55,56]. In contrast to the classical cell-based approaches that require the detergent extraction of the GPCRs out of cell membranes, the presented CF approach uses the detergent-free insertion of the nascent receptors directly into preformed ND membranes. The CF approach thus represents a

shorter, milder and less complex strategy compared to cell-based procedures, avoiding any detergent solubilization and reconstitution procedures of GPCRs that may result in receptor denaturation[57,58]. Furthermore, due to the simplified and translocon independent membrane insertion process, extensive engineering such as the deletion of terminal or internal loops is not necessary and full-length GPCRs can be synthesized[12,13,59].

Our combined strategy implementing cryo-EM, mutagenesis, signaling assays and docking studies yields insights into the molecular underpinnings of ligand binding, receptor activation and G protein coupling of the H$_2$R. Previous modelling studies using the β$_2$AR/G$_s$ complex[60] as template and histamine insertion by molecular dynamics simulations[61] proposed a salt bridge between the histamine ammonium group and residue D98$^{3.32}$ of H$_2$R, in addition to polar interactions between one imidazole nitrogen and residues D186$^{5.42}$ and T190$^{5.46}$. Our structure supports the interaction of the histamine ammonium group with D98$^{3.32}$. However, distances of the histamine imidazole ring to residues D186$^{5.42}$ and in particular to T190$^{5.46}$ are too large for polar interactions. Instead, the binding of histamine is stabilized by an H-bond interaction between the ring's Nπ atom and Y250$^{6.51}$. Besides these conserved key interactions between histamine and the conserved residues D$^{3.32}$ and Y$^{6.51}$ of histamine receptors, the divergent and less conserved residues at positions 5.42 and 5.46 in TM5 between the H$_2$R and H$_1$R were identified to participate in the ligand binding selectivity of the H$_2$R. In the H$_1$R, histamine engages in H-bond interactions with N$^{5.46}$, whereas in the H$_2$R, the agonist does not form direct contacts with the corresponding residue T$^{5.46}$ as well as the nearby H$_2$R-specific residue D$^{5.42}$. However, the latter residue seems to be important for the selective binding of H$_2$R-specific agonists, as shown by our docking studies and by site-specific mutation of D$^{5.42}$ to threonine, which specifically diminishes amthamine but not histamine signaling.

Furthermore, insights into the molecular mechanism of H$_2$R activation and on the differences in the receptor-G protein interactions between G$_s$ and G$_q$-coupled histamine receptor subtypes could be obtained. While the conserved microswitch motifs of the H$_2$R undergo similar conformational changes as in the H$_1$R, major differences are observed in the receptor-G protein coupling between the two receptors. The heterotrimeric G$_s$ protein in the H$_2$R-G$_s$ complex is shifted more towards the TM 5/6 region of the receptor in comparison to G$_q$ in the H$_1$R-G$_q$ complex. This translation results in the formation of tighter contacts of the C-terminal α5 helix of Gα$_s$ with TM5 and the TM7/H8 hinge region of the receptor and leads to a more pronounced outward displacement of TM6 compared to the H$_1$R-G$_q$ complex. Furthermore, the ICL2 of H$_2$R points deeper into the hydrophobic pocket formed between the αN, αN-β1 hinge and α5 of the Ras domain of Gα$_s$ than ICL2 of H$_1$R in the G$_q$ complex, which is engaged in more polar contacts with the C-terminal end of the αN of Gα$_q$. Moreover, as the H$_2$R-G$_s$ structure was obtained in lipid NDs, the apparent electrostatic interactions between basic residues of the αN of Gα$_s$ and the polar headgroup of the lipids might suggest a role for the G protein anchoring to the membrane to facilitate receptor coupling, as previously described[42].

In summary, the results show that our CF production pipeline can provide structural information of GPCRs with a good global resolution. We further demonstrate an alternative strategy for the functional transfer of membrane proteins from NDs to membranes of living cells[20–22]. Here, we used this approach to confirm the previously proposed homo- and heterooligomerization of the H$_1$R and H$_2$R using pulldowns and internalization assays. As the CF synthesized receptors are easily accessible for mutations or modifications, the nanotransfer strategy may also help to further characterize the GPCR interaction requirements and interfaces. We anticipate that the CF production of GPCRs will become more broadly applicable for structure determination of full-length and less engineered receptors in complex with heterotrimeric G proteins and in absence of detergents.

## Methods

### Lysate preparation
*E. coli* S30 lysates were prepared as published[54,62]. Briefly, a pre-culture containing 150 mL LB media was inoculated with *E. coli* A19 cells and incubated at 37 °C with shaking (180 rpm) over night. A fermenter was filled with 10 L YPTG medium (16 g/L peptone, 10 g/L yeast, 5 g/L NaCl, 100 mM glucose, 22 mM KH$_2$PO$_4$, 40 mM K$_2$HPO$_4$). The fermenter was inoculated with 100 mL of the pre-culture and grown at 37 °C with vigorous stirring (300 rpm) and approx. 1 mL antifoam Y-30 emulsion (Sigma-Aldrich, Taufkirchen, Germany). When an OD$_{600}$ of 3.5 to 4 was reached, the culture was rapidly cooled to 20 °C and subsequently harvested by centrifugation at 5,000 × g and 4 °C for 30 min. Cells were resuspended in 300 mL S30 buffer A (14 mM Mg(OAc)$_2$, 60 mM KCl, 6 mM β-mercaptoethanol, 10 mM Tris-acetate, pH 8.2) and centrifuged at 10,000 × g and 4 °C for 10 min. This step was repeated twice and the last centrifugation was extended to 30 min. The washed pellet was resuspended in 110 % (w/v) S30 buffer B (14 mM Mg(OAc)$_2$, 60 mM KCl, 1 mM DTT, 10 mM Tris-acetate, pH 8.2) and cells were disrupted using a French press with constant pressure of 20,000 psi. Disrupted cells were centrifuged twice at 30,000 × g and 4 °C for 30. The supernatant was adjusted to 400 mM NaCl and incubated at 42 °C for 45 min for a mRNA run-off. The solution was dialysed 2 × at 4 °C against 5 L S30 buffer C (14 mM Mg(OAc)$_2$, 60 mM KAc, 0.5 mM DTT, 10 mM Tris-acetate, pH 8.2) for 3 and 12 h, respectively. After a final centrifugation at 30,000 × g and 4 °C for 30 min, the supernatant was collected, aliquoted, flash frozen in liquid nitrogen and stored at −80 °C.

### Expression and purification of MSP1E3D1
MSP1E3D1 was synthesized in *E. coli* T7express cells (New England Biolabs, Frankfurt, Germany) and purified by taking advantage of a terminal His-tag[63]. For removal of the N-terminal His$_6$-tag, the protein was TEV digested followed by a reverse IMAC. The MSP1E3D1 solution was set to 1 mM DTT before TEV was added in a MSP1E3D1 to TEV ratio of 1:25. The mixture was dialyzed against 1 mM DTT, 0.5 mM EDTA, 50 mM Tris-HCl, pH 8.0 at 4 °C over night. Before loading on a pre-equilibrated HiTrap™ IMAC FF column (Cytiva, Munich, Germany), the mixture was centrifuged at 20,000 × g and 4 °C for 10 min. The flow through was collected and the column was washed with 10 column volumes (CVs) equilibration buffer (20 mM IMD, 100 mM NaCl, 20 mM Tris-HCl, pH 8.0). The flow through and wash fractions were concentrated to 3–5 mg/mL using Amicon ultrafiltrators (10 kDa MWCO, Merck Millipore, Darmstadt, Germany). All fractions were combined and dialyzed 2× over night at 4 °C against 5 L 10% (v/v) glycerol, 300 mM NaCl, 40 mM Tris-HCl, pH 8.0. Until ND formation, the protein was flash frozen in liquid nitrogen and stored at −80 °C.

### Nanodisc formation
Purified MSP1E3D1 was incubated with the respective lipid and supplemented with 0.1% DPC. For each lipid, NDs were assembled at defined MSP1E3D1 to lipid ratios (1:80 for DOPG, 1:80 for DOPC, 1:85 for DEPG, 1:110 for DMPG)[16,63]. Solutions were incubated at RT for 1 h with gentle stirring, before being dialyzed 3x for at least 12 h at RT against 100 mM NaCl, 10 mM Tris-HCl, pH 8.0. The mixture was centrifuged at 30,000 × g and 4 °C for 20 min. NDs were concentrated to 500–1000 μM using Centriprep concentrating units (10 kDa MWCO, Merck Millipore, Darmstadt, Germany). Lipids and DPC were purchased from Avanti Polar Lipids (Alabaster, USA).

### CF expression
Coding sequences of H$_1$R-Strep, H$_2$R-Strep and FFAR$_2$-Strep were synthesized from Twist Bioscience and cloned into vector pET29b. Constructs comprised the full-length GPCRs containing a N-terminal H-tag for improved expression[64] as well as a C-terminal Strep-tag for affinity chromatography. For the nanotransfer or determination of

expression levels in different NDs, a C-terminal mNG-fusion was introduced. CF protein expression was performed in lysates of *E. coli* strain A19 using a two-compartment configuration[62,65]. Optimal $Mg^{2+}$ concentrations were determined for each DNA template by screening within a concentration range of 14−22 mM. Analytical scale reactions were carried out in 24 well plates holding the feeding mixtures and by using Mini-CECF reactors holding 60 μL reaction mixtures. Preparative scale reactions were carried out in 3 mL Slide-A-Lyzer cassettes (10 kDa MWCO, Merck Millipore, Darmstadt, Germany) in combination with custom-made containers[66]. The reaction mixture to feeding mixture ratios were 1:17. Reactions were performed with 1 mM of each amino acid, 20 mM acetylphosphate, 20 mM phosphoenolpyruvate, 0.1 mg/mL folinic acid, 1× complete protease inhibitor (Roche, Penzberg, Germnay), 16−20 mM $Mg(OAc)_2$, 270 mM KOAc, 3 mM GSH, 1 mM GSSG, 1.2 mM ATP, 0.8 mM each CTP, GTP, UTP, and 100 mM HEPES-KOH pH 8.0. The reaction mixture in addition contained the *E. coli* lysate, 15 ng/μL DNA template, 0.3 U/μL RiboLock RNase inhibitor (ThermoScientific, Langenselbold, Germnay), 10−20 U T7 RNA polymerase, 0.04 mg/mL pyruvate kinase and 0.5 mg/mL *E. coli* tRNA (Roche, Penzberg, Germany).

For cotranslational solubilization, all GPCRs were synthesized in presence of 60 μM NDs. Reactions were incubated at 30 °C for 16 to 20 h with gentle shaking. After expression, RMs were harvested and centrifuged at 18,000 × g and 4 °C for 10 min to remove precipitates. For purification of GPCR/ND complexes, RMs were diluted 1:3 in buffer A (100 mM NaCl, 20 mM HEPES, pH 7.4). Gravity flow columns containing StrepII-Tactin resin (IBA, Goettingen, Germany) were equilibrated in buffer A and samples were loaded and re-loaded twice. The columns were washed with 10 CVs buffer A and eluted in 4 to 5 CVs [buffer A + 25 mM d-desthiobiotin]. The samples were subsequently concentrated using Amicon Ultra−0.5 mL units (MWCO 50 kDa, Millipore, Merck, Darmstadt, Germany).

### Expression and purification of Nb35
Nb35 was expressed and purified according to standard protocols[60]. Briefly, Nb35 was expressed in *Escherichia coli* BL21 cells. After lysis, it was purified using IMAC and finally subjected to size exclusion chromatography on a Superdex 200 10/300 gel filtration column (GE Healthcare) in 20 mM HEPES pH 7.5, 150 mM sodium chloride. Purified Nb35 was concentrated, flash frozen, and stored at −80 °C until further use.

### Nanotransfer
The nanotransfer is a recently developed approach and several persistent limitations must still be considered[20,21]. Nanotransfer is currently much less efficient if compared with conventional transfection, reaching approx. only 10% of a corresponding transfection efficiency. In addition, the transfer is not directed and, depending on the individual topology of the target, a significant fraction or even the majority of the transferred protein will become inserted in wrong orientation[20]. Flag-$H_1R$ and Flag-$H_2R$ were ordered from Addgene (#66400 & #66401) and de-tangonized using standard site-directed mutagenesis[67]. Fluorescence microscopy and pulldowns with transferred GPCRs were performed as described before[20]. Briefly, for fluorescence microscopy, HEK293 (ThermoFisher Scientific, Nr. R70507) cells were seeded at a density of 2 × 105 cells/well onto glass coverslips in a 12-well plate. After 24 h, fresh Dulbecco's Modified Eagle Medium (DMEM) with 0.5 μM of Strep-purified GPCR/ND complexes was added and cells were incubated for 4 h. Afterwards, they were washed 5 times with Dulbecco's Phosphate Buffered Saline. To stimulate GPCR activation and subsequent internalization, the cells were incubated with DMEM containing 100 μM histamine. After 1 h, cells were washed and fixed with RotiHistofix. For pulldowns, cells were seeded into 6-well plates and transfected with Flag-$H_1R$ and Flag-$H_2R$. The next day, 0.5 μM GPCR/ND complexes were added for 16 h. Cells

were then lysed, and anti-Strep pulldowns were performed using magnetic Strep-Tactin beads. Western blotting was performed using the Trans-Blot Turbo Transfer System (BioRad) according to the manufacturers instructions. Anti-Strep-Tactin-HRP conjugate (1:5,000; 1610381; BioRad) was used to detect Strep-tagged targets in western blots. Flag-tagged proteins were detected with an anti-Flag primary antibody (1:1,000; F3165; Sigma) and a secondary anti-mouse-HRP conjugate (1:5,000; A9917; Sigma).

### Size exclusion chromatography
SEC was carried out at 12 °C, using an Äkta purifier system (Cytiva, Munich, Germany) and Increase Superose 6 5/150 or 3.2/300 columns (Cytiva, Munich, Germany). The columns were equilibrated in sterile-filtrated, degased and pre-cooled buffer (100 mM NaCl, 20 mM HEPES, pH 7.4) before injecting the affinity purified protein samples. Flow rate was 0.15 mL/min for Superose 6 5/150 and 0.05 mL/min for Superose 6 3.2/300. UV absorbance was recorded at 280 nm and data were plotted using GraphPad Prism (v.9.5).

### Expression and purification of heterotrimeric $G_s$
$G_s$ heterotrimer was expressed and purified, as described[68]. Briefly, *Trichoplusia ni* (*T. ni*, Expression Systems Nr. 94−002 F) insect cells using baculoviruses generated by the BestBac method were used for expression of $G_s$ heterotrimer. One baculovirus encoding the human $G\alpha_s$ short splice variant and another separate baculovirus encoding both the $G\beta_1$ and $G\gamma_2$ subunits, with a histidine tag and HRV 3 C protease site inserted at the amino terminus of the β-subunit were used. *T. ni* cells were infected with the baculoviruses followed by incubation of 48 h at 27 °C. After harvest by centrifugation, cells were lysed in [10 mM Tris, pH 7.5, 100 μM $MgCl_2$, 5 M β-ME, 20 μM GDP and protease inhibitors]. The membrane fraction was collected by centrifugation and solubilized in [20 mM HEPES, pH 7.5, 100 mM NaCl, 1% Na-cholate, 0.05 % DDM, 5 mM $MgCl_2$, 5 mM β-ME, 5 mM IMD, 20 μM GDP and protease inhibitors]. After homogenization with a Dounce homogenizer, the solubilization reaction was incubated for 45 min at 4 °C. After centrifugation, the soluble fraction was loaded onto HisPur Ni-NTA resin (Thermo Scientific, Langenselbold, Germany) followed by a gradual detergent exchange into 0.1% DDM. The protein was eluted in buffer supplemented with 200 mM IMD and dialyzed overnight against [20 mM HEPES, pH 7.5, 100 mM NaCl, 0.05% DDM, 1 mM $MgCl_2$, 5 mM β-ME and 20 μM GDP] together with HRV 3 C protease to cleave off the amino- terminal $His_6$-tag. Cleaved $His_6$-tag, uncleaved fractions and 3 C protease were removed by Ni-chelated Sepharose. The cleaved G protein was dephosphorylated by lambda protein phosphatase (NEB, Frankfurt, Germany), calf intestinal phosphatase (NEB, Frankfurt, Germany), and antarctic phosphatase (NEB, Frankfurt, Germany) in the presence of 1 mM $MnCl_2$. Lipidated $G_s$ heterotrimer was isolated using a MonoQ 5/50 GL column (Cytiva, Munich, Germany). The protein was bound to the column in buffer A [20 mM HEPES, pH 7.5, 50 mM NaCl, 1 mM $MgCl_2$, 0.05% DDM, 100 μM TCEP, 20 μM GDP] and washed in buffer A. The $G_s$ heterotrimer was eluted with a linear gradient of 0 to 50% buffer B [buffer A + 1 M NaCl]. The main peak containing iso-prenylated $G_s$ heterotrimer was collected and dialyzed against [20 mM HEPES, pH 7.5, 100 mM NaCl, 0.02% DDM, 100 μM TCEP and 20 μM GDP]. The protein was concentrated to 250 μM, 20 % glycerol was added and the protein was flash frozen in liquid nitrogen and stored at −80 °C until use.

### Cryo-EM sample preparation
The $H_2R$/ND/$G_s$/Nb35-His complex was formed co-translationally. Reactions were supplemented with final concentrations of 15 ng/μL $H_2R$-Strep template, 60 μM NDs (DOPG) without His-tag, 10 μM purified $G_s$ heterotrimer and 15 μM Nb35-His. Reactions were incubated for 16 h at 30 °C with gentle shaking. Samples were harvested by centrifugation at 18,000 × g and 4 °C for 10 min and subsequently

incubated with 1 U/µL apyrase on ice for 90 min. The H$_2$R/ND/G$_s$/Nb35-His complex was purified by IMAC at 4 °C. Samples were diluted 1:3 in IMAC buffer A [100 mM NaCl, 20 mM HEPES, pH 7.4] and loaded and reloaded twice on a pre-equilibrated gravity-flow IMAC column. The column was washed with 4 column volumes IMAC buffer A and 4 column volumes IMAC buffer B (IMAC buffer A + 30 mM IMD). Samples were eluted using IMAC elution buffer (IMAC buffer A + 300 mM IMD) and concentrated using Amicon Ultra−0.5 mL units (50 kDa MWCO, Merck Millipore, Darmstadt, Germany). SEC was performed as described above. Complex containing fractions were pooled and concentrated again using Amicon Ultra−0.5 mL units (50 kDa MWCO, Merck Millipore, Darmstadt, Germany).

## Cryo-EM data acquisition

Two datasets were collected using the same settings, but different concentrations. Samples were prepared as previously described[69]. Specifically, the sample was concentrated to 1.4 mg/ml for the first and 2.8 mg/ml for the second dataset. C-flat grids (Protochips; CF-1.2/1.3−3Cu-50) were prepared by glow-discharging them with a PELCO easiGlow at 15 mA for a duration of 45 s. A total of 3 µL of the sample were promptly applied to the grids and immediately plunge-frozen in liquid ethane with the use of a vitro bot Mark IV (Thermo Fischer). This process was conducted at 4 °C with a relative humidity of 100%. Data was collected on a Glacios microscope (Thermo Fischer), operating at 200 kV and equipped with a Selectris energy filter (Thermo Fischer) with a slit with of 10 eV. Movies were recorded using a Falcon 4 direct electron detector (Thermo Fischer) at a nominal magnification of 130,000 which is equal to a calibrated pixel size of 0.924 Å per pixel. The dose rate for collection was set to 5.22 e- per pixel with a total dose of 50 e- per Å$^2$. 46896 movies were automatically gathered using EPU software (v.2.9, Thermo Fischer) with a defocus range of −0.8 µm to −2.0 µm and stored in EER (electron-event representation) format.

## Cryo-EM image processing

The dataset was processed using cryoSPARC (v.4) (Supplementary Fig. 4). The preprocessing of the movies entailed patch-based motion correction, patch-based CTF estimation and filtering based on the CTF fit estimates using a cutoff at 5 Å. This resulted in a remaining dataset of 41594 micrographs.

Distinct 2D classes were identified and then utilized for further template-based particle picking. This process yielded a collection of 30.5 million particles. The particles were extracted within a box size of 288 pixels and then Fourier cropped to 72 pixels. Subsequent rounds of 2D classification were applied to refine the stack further, resulting in 1.5 M particles. These particles were then used for ab-initio 3D reconstruction. Particles from the top four reconstructions were merged and further refined by heterogeneous refinement, followed by another round of ab-initio 3D reconstruction. A final stack consisting of 425 thousand particles was utilized for a Non-Uniform refinement, resulting in a consensus map with a resolution of 3.5 Å. To enhance the quality of the map further, a local refinement was executed, resulting in a final map with a resolution of 3.4 Å.

## Model building and refinement

The preliminary atomic structure was computed using ModelAngelo (v.0.3). Subsequently, the generated structure was manually inspected in Coot (v.0.9) and iteratively refined using *phenix.real_space_refine* within Phenix (v.1.19). Further enhancement of the density map's quality was accomplished through *phenix.density_modification*. Validation reports were generated by MolProbity. Alignment of the experimental structure was performed with a model generated by the AlphaFold2 (AF2)[31] multimer tool via the COSMIC2 platform[32] using the UCSF Chimera tool "MatchMaker" (v. 1.16). The final density map and corresponding atomic structure have been deposited in the Electron Microscopy Data Bank and the Protein Data Bank. These submissions can be found with the PDB ID 8POK and the EMDB ID 17793, respectively. All structural data was visualized using ChimeraX and Pymol (v.4.3).

## Docking calculations

The reported H$_2$R structure, as well as the H$_1$R structure in an active conformation (PDB ID 7DFL), were prepared for docking calculations by adding protons and consequent energy minimization, acylation of the N-terminus, N-methylation of the C-terminus, and conversion of the mutated back to the wild-type or construction of the unmodeled residues using the Molecular Operating Environment (MOE, v.2022.02). The predicted binding modes of the compounds were energy-minimized using the MMFF94x force field. Binding pocket residues were relaxed through energy minimization in presence of the different compounds using the AMBER force field, as implemented in MOE.

The following softwares were used for docking calculations: Autodock Vina (v.1.1.2)[70], DOCK3.7[71], the OpenEye programs FRED (v.3.3.0.3) and HYBRID (v.3.3.0.3)[72], and SEED (Solvation Energy for Exhaustive Docking) (v.4.0.0)[73]. The differences among these softwares lie in both the search algorithms used to sample the different orientations of the ligands in the binding pockets and the energy terms with which these orientations are scored. Autodock Vina applies a Lamarckian genetic algorithm to generate and optimize the possible orientations, thus accounting for ligand flexibility. In contrast, DOCK uses a shape matching method to sample the different ligand conformations, thus treating them as individual rigid bodies. OpenEye programs FRED and HYBRID also consider each ligand's flexibility by pre-generating different conformations using the program OMEGA. HYBRID additionally uses information from the experimental pose of the ligand, when available. Scoring functions are different for each of the employed softwares: A knowledge-based scoring function is applied by Autodock Vina for the ranking of poses, while force field-based and empirical functions are applied by DOCK and OpenEye softwares, respectively. The SEED software, differently from all the previously mentioned ones, is used for docking of fragments, and similar to DOCK, its search algorithm is based on exhaustive matching of conformers to the binding region. It also uses a force field-based scoring function, which gives particular emphasis to protein and fragment desolvation upon binding.

UCSF Chimera[74] was used for generating the pdbqt format file used in Autodock Vina. The top-ranked binding modes from each docking calculation were inspected visually and those that had unfavorable interactions not considered further. RDKit software (v.2018.9.3) was used for RMSD calculations (RDKit: Open-source cheminformatics. https://www.rdkit.org). Pymol (v.4.3) was used for visualization and pose evaluation.

## Cell culture & transient transfection for surface ELISA and BRET assays

Wildtype and mutant H$_{1/2}$R and the G protein biosensors, G$_q$-CASE and G$_s$-CASE (plasmids are available from Addgene (https://www.addgene.org/browse/article/28216239/)[37], were transiently expressed in HEK293A cells grown in Dulbecco's Modified Eagle's Medium (DMEM) supplemented with 2 mM glutamine, 10% fetal calf serum, 0.1 mg/mL streptomycin, and 100 units/mL penicillin at 37 °C with 5% CO$_2$. Each mL of resuspended cells (300,000 cells/mL) was mixed with a total of 1 µg DNA and 3 mL PEI solution (1 mg/mL) (Merck KGaA, Darmstadt, Germany). The total DNA mix of 1 µg was composed of 0 - 500 ng receptor and 500 ng G-CASE plasmid. Empty vector (pcDNA) was used to compensate for smaller amounts of receptor DNA. 100 µL transfected cells were seeded per well onto 96-well plates (Brand, Wertheim, Germany) and grown for 48 h at 37 °C with 5% CO$_2$. White plates were used for BRET experiments and transparent, flat bottom 96-well plates (Brand, Wertheim, Germany) were used for the assessment of receptor surface

levels. Absence of mycoplasma contamination was routinely confirmed by PCR.

## Assessment of receptor surface expression through live-cell ELISA

To quantify cell surface receptor expression, HEK293 (ThermoFisher Scientific, Nr. R70507) cells transfected with G-CASE and pcDNA or N-terminally FLAG-tagged $H_{1/2}R$ constructs were grown for 48 h in transparent 96-well plates (Brand, Wertheim, Germany) and washed once with 0.5% BSA (Merck KGaA, Darmstadt, Germany) in PBS. Next, cells were incubated with a rabbit anti-FLAG M2 antibody (142 ng/mL) (Cell Signaling Technology, Danvers, MA, USA) in 1% BSA–PBS for 1 h at 4 °C. Following incubation, the cells were washed three times with 0.5% BSA–PBS and incubated with a horseradish peroxidase-conjugated goat anti-rabbit antibody (30 ng/ml) (Cell Signaling Technology, Danvers, MA, USA) in 1% BSA–PBS for 1 h at 4 °C. The cells were washed three times with 0.5% BSA/PBS, and 50 μl of the 3, 3', 5, 5' tetramethyl benzidine (TMB) substrate (BioLegend, San Diego, CA, USA) was added. Subsequently, the cells were incubated for 30 min and 50 μl of 2 M HCl was added. The absorbance was read at 450 nm using a BMG ClarioStar Plus plate reader.

## BRET-based G protein activation experiments

G protein activation experiments with the G-CASE biosensors were conducted as previously described (Reference to PMID: 34516756). Briefly, transfected cells grown for 48 h in 96-well plates were washed with Hanks' Balanced Salt solution (HBSS) and incubated with 1/1,000 dilution of furimazine stock solution (Promega, WI, USA). After incubation for 2 min, BRET was measured in three consecutive reads followed by addition of agonist (histamine/amthamine) solutions or vehicle control and subsequent BRET reads. All experiments were conducted at 37 °C. Nluc emission intensity was selected using a 470/80 nm monochromator and cpVenus emission using a 530/30 nm monochromator in a CLARIOstar plate reader with an integration time of 0.3 s.

## BRET data analysis

BRET ratios were defined as acceptor emission/donor emission. The basal BRET ratio before ligand stimulation ($BRET_{basal}$) was defined as the average of at least three consecutive reads. To quantify ligand-induced changes, D BRET was calculated for each well as a percent over basal $[(BRET_{stim} - BRET_{basal})/BRET_{basal}] \times 100)$. Next, the average ΔBRET of vehicle control was subtracted. Data were analysed using GraphPad Prism v.9.5 software (GraphPad, San Diego, CA, USA). Data from BRET concentration–response experiments were fitted using a three-parameter fit. Data from cell surface ELISA experiments were corrected for background by subtracting the values obtained for pcDNA-transfected cells.

## Reporting summary

Further information on research design is available in the Nature Portfolio Reporting Summary linked to this article.

## Data availability

The following datasets has been used for structural analysis and comparison: PDB 7DFL, PDB 7UL3, PDB 3SN6, PDB 7BZ2, PDB 7DHI, PDB 7DHR, and PDB 7F1Z. The EM map for the complete $H_2R$ molecule has been deposited in the EMDB under accession code EMD-17793. Atomic coordinates for $H_2R$ have been deposited in the Protein Data Bank under the accession code PDB 8POK. Source data are provided with this paper.

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

## Acknowledgements
We thank Birgit Schäfer for helpful discussions and technical assistance. We are further grateful to Ulrich Ermler for valuable help in structure calculation, to Betsy White for help with G protein expression and purifications and to Juliane Bernhard for artwork. The work was funded by the DFG project BE1911/9-1 (F.B.), by the Center for Biomolecular Magnetic Resonance (V.D., F.B.) and by the LOEWE project GLUE of the state of Hessen. LOEWE GLUE financed the theses of Z.K., S.U., and M.P. Financial support was further obtained by the Fonds der Chemischen Industrie (SK-208/16).

## Author contributions
Sample preparations and biochemical studies were done by Z.K. and D.H. K.S., D.J., K.P., and A.M. performed cryo-EM data acquisition and analysis. D.H. provided heteromeric G-proteins. S.U. performed the nanotransfer and analysis in HEK293T cells. H.S., D.H., and S.P. characterized mutants and analyzed ligand selectivity. M.P. and P.K. performed molecular docking studies. V.D. provided essential support and infrastructure. F.B. and D.H. conceived the project. All authors contributed to manuscript writing, data analysis, reading and approving the final version of the manuscript.

## Funding

## Competing interests
The authors declare no competing interests.
