## [Peer Review File · Nature Communications]

Cryo-EM structure of cell-free synthesized human histamine 2 receptor/Gs complex in nanodisc environmentREVIEWER COMMENTS

Reviewer #1 (Remarks to the Author):

This manuscript by Köck et al. describes a pipeline for cell-free expression in *E. coli* A19 lysates of the histamine H2 receptor directly into DOPG-containing MSP nanodiscs, which allowed the authors to avoid exposure of the receptor to detergents and maintain a more “native-like” lipid environment during expression and purification. The purified receptor was introduced into live cells using a nanotransfer technique for pull-down experiments to establish its retained ability to form homodimers, as well as hetero-dimers with differentially tagged H1R, establishing that it remains functional. The purified protein was also used successfully to enable the first cryo-EM structure of the H2R receptor in its active, histamine-bound state in complex with the G α s complex, stabilized by Nb35. The authors compared their structure to the inactive-state structure of H2R and structures of H1R to elucidate the mechanism of activation and to identify residues underlying differences in ligand selectivity, G protein binding mode, and agonist binding selectivity, the latter through docking simulations and mutagenesis experiments. The manuscript describes a novel application of cell-free expression to synthesize a GPCR directly into nanodiscs and represents a well-executed structural study of the histamine H2 receptor. However, it does not provide the level of technical or mechanistic advance typical of other papers published in *Nature Communications* and thus may be better suited to publication in a more subject-specific journal. Specific comments follow.

1. The primary advancement described in this manuscript is the direct cell-free expression of a GPCR into pre-formed nanodiscs, avoiding exposure of the protein to detergent. Why was the *E. coli* system chosen over alternative eukaryotic cell-free expression systems? Since many receptors are post-translationally modified, this system could have significant functional impacts, limiting how broadly applicable this system would be to other eukaryotic receptors. Does the lipid environment of the nanodisc impact the H2R structure or function in any observable way that makes this system superior to eukaryotic cell-free expression or detergent extraction?

2. The authors claim the first structure of a CF-synthesized GPCR and the first GPCR/Gs complex obtained in lipids. However, they tested only versions of phosphatidylglycerol (PG), cardiolipin (CL), and phosphatidylcholine (PC) during nanodisc optimization. The reason for these choices is not clear, since PC is a neutral phospholipid not present in *E. coli*, while CL is restricted to mitochondria in animals and PG is an acidic phospholipid that is altogether very rare in animal cells, where the receptor is natively expressed. This makes the lipid environment of the purified receptor quite non-native and thus it's difficult to draw conclusions about mechanistic or functional effect of the lipids in these experiments. It would be worthwhile to test this system with nanodiscs containing other neutral and acidic phospholipids found in human membranes, such as PE, PS, PI, and/or phosphoinositides.

3. How might PG have affected the receptor's structure and function? The authors used nanotransfer into cells to evaluate the receptor's functionality and ability to homo/heterodimerize. However, following nanotransfer, the receptor would be in a more physiologically relevant membrane environment compared to that of the nanodisc. Thus, any artifacts of the nanodisc lipid composition that might have impacted the structure and function of the receptor in the nanodisc could have been "erased" by re-introduction to cells. Did the authors observe any evidence of lipids in their structure that might have impacted the receptor structure?

4. As the authors state, "The lipid composition of ND membranes can be of crucial importance of the efficiency of cotranslational membrane insertion as well as for the function of the inserted membrane proteins." (Lines 74-76) Can the authors comment on why DOPG and DEPG were most efficient for H2R insertion? Is this phenomenon related to the E. coli CF expression system as opposed to the requirements of the H2R receptor? Do other membrane proteins expressed in this system exhibit a similar preference for versions of PG? If so, this could be a major limitation of this system for expression of other receptors in a functional state in similar E. coli cell-free expression systems.

5. The significance of potential Gas interaction with POPG lipids is unclear, since the receptor is in a non-native lipid environment. However, a similar type of interaction has been extensively documented for other receptors. As mentioned above, other more human-relevant acidic phospholipids would have been worthwhile to test.

6. The authors used histamine-containing H1R and H2R structures to evaluate binding mode prediction tools, settling on Autodock Vina. They then used Autodock Vina to predict binding modes for H2R-selective agonists in both receptors to understand selectivity and validated predictions by swapping receptor-specific residues predicted to be involved in agonist interaction. While this approach has provided insights into receptor agonist selectivity, more convincing insights could have been obtained by using their CF expression system to enable additional structures with these agonists. Did the authors pursue structures with other agonists, like amphetamine, using their CF expression pipeline?

Reviewer #2 (Remarks to the Author):

The submitted article entitled "Cryo-EM structure of cell-free synthesized human histamine H2 1 receptor coupled to heterotrimeric Gs protein in lipid nanodisc environment", by Kock1 et al., presents a tour de force approach and analysis on the structure and function of human histamine 2 receptor. The manuscript is highly focused and well written. The techniques and results are sequentially presented and easy for the reader to comprehend. Overall, the article is extremely well written, and the authors have

provided detailed information, protocols, and statistical analysis of the data. All the figures reflect the authors findings and conclusions.

First, the authors show that cell-free expression is a valuable tool for screening lipids and other additives such as lipids, nanodisc and G-protein complexes. The cell-free lysates are then scaled up for production to provide material for biochemical characterization and novel structural insight into the H2 type receptors bound with ligand and G-protein. Furthermore, the authors show that this approach does not require a detergent solubilization step for the GPCRs and allowed for full length expression of the native like amino acid sequence without the need for stabilization domains/mutants. The authors were also able to provide new data on the complex at a resolution of 3.4 angstrom, which is comparable to analogous structures of GPCR complexes expressed and purified from eukaryotic cell-based systems but relied on other techniques. Regarding these other structures, the authors may have missed an opportunity to highlight the uniqueness of this approach. To biochemically validate the predicted complexes, the authors used mutagenesis for comparative studies to WT in the presence of different ligands along with G-protein activation. This approach also confirmed homo- and hetero-oligomerization as well as the impact of specific side chains on receptor function. The nanodelivery technique and data could have been exploited to provide greater information on correctly folded and functional complexes.

I believe this is an important area of research in the GPCR field and the general readers will be extremely interested in the manuscript and its findings related to novel techniques for producing membrane proteins in multiprotein complexes that are correctly folded and functional. I would recommend this paper for publication after addressing several minor specific comments (see below).

Specific Comments:

1. It was not clear what the authors meant by "...semi-defined CF environment obtained from E. coli strain A19...? Was this referring to the lysate itself, the buffer conditions or some genetic alteration in the A19 strain?

2. The authors use the term "co-translation" multiple times in the article, but is never clear to the reader when two or more things are being co-translated?

3. The following papers should be considered for citation given they demonstrated the use of preformed nanodisc for generating membrane proteins such as GPCRs (Katzen, F., et al. (2008). "Insertion of membrane proteins into discoidal membranes using a cell-free protein expression approach. *Proteome Res* 7(8): 3535–42.)). These same authors also demonstrated this same technique for producing full-length GPCRs, but required insertion of a T4L stabilizing domain for the GPCR of interest (Yang et al. Cell-free synthesis of a functional G protein-coupled receptor complexed with nanometer scale bilayer discs.

BMC Biotechnol 11, 57 (2011). <https://doi.org/10.1186/1472-6750-11-57>). In my opinion, the authors should emphasize the benefit of their approach compared to previous publications that required the use of the stabilizing insertions for structural studies. How does the current manuscripts resolution compare to previously solved crystal structures?

4. Did the authors consider the opportunity to use computational structural methods for comparative studies to predicted GPCR structures? There should have been a wealth of information validating or disproving the predictive structures in databases such as AlphaFold regarding the human histamine 2 receptor.

Reviewer #3 (Remarks to the Author):

The manuscript by Köck Z et al. describes the cryo-electron microscopy (cryoEM) three-dimensional (3D) structure of the active human histamine H2 receptor (H2R) in complex with its natural ligand histamine, the heterotrimeric Gs protein, and the stabilizing nanobody Nb35. It should be emphasized that this is the first structure of a G protein-coupled receptor (GPCR)/G protein signaling complex in which the GPCR is produced using a cell-free system (E. coli lysate) supplemented with purified Gs protein and Nb35. The quantities of H2 receptor are compatible with structural analysis. The study enabled single-particle analysis of the complex at a global resolution of 3.4 Å allowing to provide a structural basis for the activation and ligand recognition of this histamine H2 receptor which plays a central role in diverse pathophysiological situations. Again, the H2R is cotranslationally inserted into preformed nanodiscs (ND), and it is demonstrated it can be directly transferred from these particles to living cell (HEK293 cells) plasma membranes. The transfer is efficient since it is shown the receptor is internalized upon ligand binding and is able to form heterodimers with transfected histamine H1 receptors. Finally, the determination of the H2R/Gs/Nb35 complex structure allowed the authors to understand what makes the specificity of several H2-selective ligands to H2R, relative to other histamine receptors.

This work is a nice combination of cell-free GPCR production, cryoEM structure analysis, nanotransfer of GPCR from ND particles to living cells, mutagenesis, signaling assays, and ligand docking approaches. This strategy should become in the future more broadly applicable to other GPCRs or other membrane protein families. Regarding the H2R, it may help in the rational design and development of novel H2R-selective ligands for treating gastric acid secretion, cardiac activity and vasodilatation.

The expertise of the authors in the scientific field of GPCRs and molecular pharmacology is internationally well renowned, the methods developed are state-of-the-art and the results presented in the article are really interesting for GPCR's research community. However, I have some concerns and also some minor comments before the manuscript would be accepted for publication in Nature Communications.

1/ major comments:

a) My first criticism is focused on the mutagenesis study and more particularly on assessment of receptor expression levels to analyze BRET-based G protein dissociation (which means activation in that case). It is not sufficiently described, neither in methods nor in figure legends (Fig 4b and c, Fig 6c and d, Fig S7). I understood the expression level of some mutants is very low and that of wild-type (WT) has to be adjusted to compare ligand-induced G protein activation in a rigorous way. Results are shown as dose-response curves of Gs or Gq activation depending on ligand concentration and are measured as deltaBRET. This is OK. But, for instance, in the panels b and c of figure 4, 50% or 10% of WT (H1R or H2R) are used in order to compare correctly receptor expression levels between wild-type and mutants. What does mean 50%, 10%? What is the 100%? Unfortunately, Fig S7 does not help to better understand these values. What are the quantities of plasmid transfected for each WT and mutant receptor to manage comparable levels of expression? This should be indicated in Methods.

Moreover, in Figure 4, the H2R mutant T190N is analyzed whereas in Fig 6 it is explained that it is not included because of its very low expression level. Thus, is this mutant correctly expressed or not? This is not clear.

b) My second criticism is about nanotransfer of H2R into living cells from the nanodiscs to the plasma membrane of these cells. The strategy is strikingly interesting and promising in the case of GPCRs or other membrane proteins which are difficult to endogenously express through transfection of DNA. In the present study, efficient transfer of the nanodisc-embedded receptor into plasma membranes of HEK293 cells is demonstrated through ligand-induced internalization (measured with fluorescence of monomeric neongreen protein fused at the C-terminus of H2R). Colocalization with transfected H1R is also shown, as well as heterodimerization with this related histamine receptor.

To my opinion, at least two assays should be added to complete the study. To show that the H2R-transferred is totally functional and possess equivalent properties to H2R synthesized in mammalian cells, a comparison with a transfected H2R may be done. In other words, determination of the Kd of histamine for the H2R with cells expressing H2R after transfection or expressing H2R after nanodisc-membrane transfer could be performed using ligand binding assays (radioactivity or FRET approaches). In addition, accumulation of cAMP upon histamine binding (dose-response curves) to HEK293 cells transfected with H2R plasmid or transferred with H2R/ND particles should be compared to check whether the EC50 values are in the same range.

c) The third major criticism deals with ligand docking calculations. The comparison of the reported cryoEM H2R (in complex with histamine) structure with that of H1R (also in complex with histamine) already available gives very important information to understand differences in histamine docking within the two related receptor orthosteric pockets, and more importantly to determine what makes the selectivity of H2-selective compounds for H2R. Indeed, the authors demonstrated that divergent residues at positions 5.42 and 5.46 (Ballesteros-Weinstein nomenclature) in TM5 were identified to participate in the agonist binding selectivity of the H2R.

The docking procedures are more questionable, in particular the SEED predictions. Indeed, I do not understand the different pictures of Fig S9a. First the legend of this figure is too short and does not help in complete interpretation of the data. If the legend of panel a is equivalent to that of panel b, this would mean that predicted binding poses are colored in sand whether those which are experimentally resolved are in orange. However, there is only one histamine binding pose described in the 3D structure of the histamine-H2R/Gs/Nb35 complex, not 6, even not 2 as indicated in the legend (top-2-scored binding modes are reproducing the experimentally resolved orientations). Do the different pictures in Fig 9a correspond to different iterations of the docking molecular simulations? Or do they correspond to different clusters of binding poses?

I would remove the Figure S9 panel a, which is not necessary, taking account all other data presented in the manuscript. I would keep only the AutoVina docking predictions. In addition, Table S2 and S3 should have captions to explain the different steps in the docking procedures and calculation of binding energy constants.

2/ minor comments: there are many.

a) Is the monomeric neongreen (NG) fluorescent protein still present in the H2R construct for cryoEM analysis of the particles? Palmitoylation of cysteine 305 is not performed in the cell-free production system, leading probably to a disordered receptor C-terminus. Would the presence of the mNG fluorescent module participate in this increased flexibility of the protein and to conformational heterogeneity? This should be discussed.

b) Page 5, line 211: please write 7UL3 instead of 7IL3.

c) Is the affinity of histamine equivalent for H1R and H2R? it should be indicated in the paragraph "comparison of histamine binding between the H2R and H1R". References should be added.

d) Page 7, second paragraph (lines 311-323): the Extended Data Figure S10D should be referred somewhere in this paragraph.

e) Page 8, line 356: please replace Fig. 7C with Fig8C.

f) Fig S1: in panel a, insertion of H2R into nanodiscs with different lipids is shown and its concentration measured using fluorescence of mNG. Does this mean that in some conditions, half of the preformed discs can insert the synthesized H2R (30 uM of receptor versus 60 uM of ND)? What means c in c(H2R-mNG) [uM]? In panel b, it would be interesting to show a SEC profile of empty nanodiscs for comparison, at least that with DOPG, since this lipid has been chosen for subsequent studies (DMPG is also interesting since the fraction of proposed folded H2R seems to be well isolated). Moreover, relative absorbance is indicated in the y axis: is the value different for each H2R/ND complex? It is not indicated. This would be useful for comparing the yields of H2R/ND purification. I have the same remark for panel

c. At the end of the legend, void volume is abbreviated by V. In the corresponding figure, V is not mentioned. I guess V is indicated with the arrow.

g) Fig S5: the density of TM7 is shown on the left. Corresponding numbering indicates amino-acids from 267 to 323. This would mean a density for TM7-helix8 and part of the C-terminus. This is not possible. Please correct.

h) Fig S6: in the sequence alignment, asterisks indicate cysteines involved in disulfide bridges. Cysteine 246 of H2R is not involved in a disulfide bridge with cysteine 174. Please put an asterisk for cysteine 91 instead of cysteine 246.

i) Fig S9: In the title of this figure, please remove "and BRET assays", since the figure only deals with histamine docking poses.

We thank the reviewers for their time, their careful evaluation of our manuscript and their helpful and constructive questions. We provide a point-by-point response below.

Reviewer #1 (Remarks to the Author)

This manuscript by Köck et al. describes a pipeline for cell-free expression in *E. coli* A19 lysates of the histamine H2 receptor directly into DOPG-containing MSP nanodiscs, which allowed the authors to avoid exposure of the receptor to detergents and maintain a more “native-like” lipid environment during expression and purification. The purified receptor was introduced into live cells using a nanotransfer technique for pull-down experiments to establish its retained ability to form homodimers, as well as hetero-dimers with differentially tagged H1R, establishing that it remains functional. The purified protein was also used successfully to enable the first cryo-EM structure of the H2R receptor in its active, histamine-bound state in complex with the Gas complex, stabilized by Nb35. The authors compared their structure to the inactive-state structure of H2R and structures of H1R to elucidate the mechanism of activation and to identify residues underlying differences in ligand selectivity, G protein binding mode, and agonist binding selectivity, the latter through docking simulations and mutagenesis experiments. The manuscript describes a novel application of cell-free expression to synthesize a GPCR directly into nanodiscs and represents a well-executed structural study of the histamine H2 receptor. However, it does not provide the level of technical or mechanistic advance typical of other papers published in Nature Communications and thus may be better suited to publication in a more subject-specific journal. Specific comments follow.

Q: *1. The primary advancement described in this manuscript is the direct cell-free expression of a GPCR into pre-formed nanodiscs, avoiding exposure of the protein to detergent. Why was the E. coli system chosen over alternative eukaryotic cell-free expression systems? Since many receptors are post-translationally modified, this system could have significant functional impacts, limiting how broadly applicable this system would be to other eukaryotic receptors. Does the lipid environment of the nanodisc impact the H2R structure or function in any observable way that makes this system superior to eukaryotic cell-free expression or detergent extraction?*

A: Eukaryotic cell-free expression systems are in our hands by far not as efficient as systems based on *E. coli* lysates and we are not aware of any publication that shows the purification of GPCR/G protein complexes from eukaryotic cell-free systems. In fact, the presented H2R complex is the first structure of a GPCR synthesized by a cell-free system. Furthermore, eukaryotic lysates are much more cost intensive, difficult to prepare and with a high batch-to-batch variation in quality.

Posttranslational modifications of microsome-containing eukaryotic lysates are often claimed to be a benefit. However, we could show in a previous publication that PTM machineries in insect cell-free lysates are rapidly titrated out if the expression rate of the membrane protein is increased above a few micrograms per mL (Merk et al., 2015). Upon preparative scale expression, which is essential for structural approaches, only a small fraction of the synthesized membrane protein is thus glycosylated and a heterogeneous sample will be produced. A further concern is the quality of glycosylation in eukaryotic cell lysates. In intact cells, glycosylation is a sequential process performed by the concerted action of specific glycosyl transferases and used as trafficking signal to guide the membrane protein through the various membrane compartments of the cell. In lysates of disrupted cells, microsomes originating from various membrane compartments may have different glycosylation systems resulting into heterogeneous patterns. Glycosylations are completely absent in the *E. coli* cell-free expression system, while disulfide bridge formation can easily be fine-tuned by modulating the Redox conditions.

Notably, deletion of the N-terminal glycosylation sites in the canine histamine H2 receptor (by mutagenesis or tunicamycin treatment) has been shown **not** to affect the ligand binding affinity for the antagonist tiotidine or the endogenous agonist histamine (Fukushima et al., 1995, *Biochem J.*). Similarly, the deletion of the palmitoylation site in H8 of the H2R showed comparable ligand binding and functional coupling of the receptor to adenylate cyclase stimulation (Fukushima et al., 2001, *Biochimica et Biophysica Acta*).

Furthermore, eukaryotic cell lysates containing microsomes are furthermore hardly suitable for the implementation of nanodiscs. We showed in a recent publication that nanodiscs rapidly fuse with cell membranes and disintegrate (Umbach et al., 2022, *Front. Bioeng. Biotechnol.*). Membrane proteins in eukaryotic cell lysates have therefore to be synthesized by insertion into the microsomal membranes. However, this will give no advantage over conventional expression systems using living cell systems such as Sf9, as classical detergent extraction has then to be applied as well.

Detergent micelles are completely artificial environments while the topology of lipid bilayers in nanoparticles provide membrane thickness, fluidity, and lateral pressure similar as in natural membranes and they are thus considered to be advantageous for structural analysis (e.g. Mio & Sato 2018). As no structure of H2R in active conformation is available, we cannot judge whether the nanodisc environment of our complex shows differences to a corresponding structure in detergent. However, nanodiscs assembled with PC or PG lipids are an established and widely used key tool for membrane protein research and numerous structural and functional reports of various membrane proteins inserted into nanodiscs have been published before (e.g., Kofuku et al., 2014; Whorton et al., 2007; Staus et al., 2019; Zhang et al., 2021; Inagaki et al., 2012; Thakur et al., 2023). A comparison of the D2R-Gi structure obtained in NDs (POPC/POPG/cholesterol) (PDB ID 6VMS) with a structure determined in detergent (0.00075% LMNG, 0.00025% GDN, 0.00015% CHS) (PDB ID 7JVR) demonstrates that the structures of the receptors alone and the complexes are very similar with RMSD values of 0.697 Å and 0.745 Å, respectively. The same is true for the structures of the NTSR1 obtained in NDs (POPC/POPG) (PDB ID 7LOP) vs detergents (0.00075% LMNG, 0.000075% CHS, 0.00025% GDN) (PDB ID 6OS9) (RMSDs of 0.813 Å and 1.149 Å for the receptor alone or the entire complex, respectively), suggesting that both high concentration of PG lipids and detergent micelles result into similar structures of GPCRs and their signaling complexes.

Advantages of the established process using *E. coli* lysates are the reduced complexity of membrane protein synthesis and trafficking, generating a high degree of operational control. As an important consequence, full-length GPCRs can be synthesized in contrast to the vast majority of GPCR structures originating from cell-based expression systems that represent heavily engineered GPCR derivatives containing deletions of both termini and IL3, as well as large insertions such as T4 lysozyme. Our process further significantly streamlines the cryo-EM sample preparation of GPCR complexes and exclusively allows to synthesize detergent sensitive targets by eliminating any extraction and reconstitution steps.

Q: 2. *The authors claim the first structure of a CF-synthesized GPCR and the first GPCR/Gs complex obtained in lipids. However, they tested only versions of phosphatidylglycerol (PG), cardiolipin (CL), and phosphatidylcholine (PC) during nanodisc optimization. The reason for these choices is not clear, since PC is a neutral phospholipid not present in E. coli, while CL is restricted to mitochondria in animals and PG is an acidic phospholipid that is altogether very rare in animal cells, where the receptor is natively expressed. This makes the lipid environment of the purified receptor quite non-native and thus it's difficult to draw conclusions about mechanistic or functional effect of the lipids in these experiments. It would be worthwhile to test this system with nanodiscs containing other neutral and acidic phospholipids found in human membranes, such as PE, PS, PI, and/or phosphoinositides.*

A: Nanodiscs are commonly assembled with PC or PG lipids and many studies on GPCRs have been performed in this lipid environment (e.g., Kofuku et al., 2014; Whorton et al., 2007; Staus et al., 2019; Zhang et al., 2021; Inagaki et al., 2012; Thakur et al., 2023). It should furthermore be considered that the lipid composition of eukaryotic membranes is highly variable and not static in composition and stoichiometry. Thousands of different lipid and protein components of eukaryotic membrane do exist. Membrane proteins can furthermore have a very specific membrane microenvironment, which is unfortunately not known for H₂R and most other GPCRs. The current state of the art thus does not allow to define a membrane typical for a natural H₂R environment. However, as a significant step towards that direction we solubilized the GPCR complex in a more native-like general lipid bilayer environment instead of using the completely artificial micelle environment that is mostly been used for structural studies of GPCRs

PC is very common in eukaryotes and was selected because H₂R is a human protein. Negatively charged lipids such as PG increase the efficiency of translocon-independent membrane integration (Ridder et al., 2001). A similar effect is described for cardiolipin which increases membrane fluidity (Unsay et al., 2013) and this lipid was thus tested whether it improves the H₂R membrane insertion efficiency. We added a better explanation for the lipid selection including the mentioned references in the text (Lanes 88-99):

“Full-length human H₂R was cotranslationally inserted into supplied preformed NDs by CF expression. Lipid type and charge can influence the membrane insertion efficiency and subsequent folding of a nascent membrane protein (Henrich et al., 2016, Köck et al., 2022). Therefore, initial expression screens were performed with a H₂R-mNG derivative and a set of NDs assembled with different lipids, and the resulting mNG fluorescence in the supernatant was analyzed as a measure for the overall H₂R-mNG solubilization. The screen included PC lipids, common in eukaryotes, and negatively charged PG lipids due to their potential to enhance the efficiency of translocon independent membrane integration (Ridder et al., 2001). A similar effect has been described for cardiolipin, which increases membrane fluidity and may therefore also affect membrane insertion of H₂R (Unsay et al., 2013). In addition, the effect of cholesterol was analyzed, as it is able to stabilize some GPCRs. Cardiolipin and cholesterol do not form ordered bilayers and were thus analyzed as additive in DOPG membranes.”

Some other proposed lipids such as PS or PI do not form pure bilayer but could be added in a certain ratio into nanodisc membranes. Since a more detailed knowledge of a H₂R/lipid interaction in cells is missing, we focused on improving the membrane insertion efficiency and the apparent homogeneity of the H₂R complex. Considering the high diversity of possibly interacting cellular membrane components in combination with the lack of any cell-based evidences, any lipid mixture in the nanodisc membrane would currently be pure speculation and obtained results of limited value.

Q: 3. *How might PG have affected the receptor's structure and function? The authors used nanotransfer into cells to evaluate the receptor's functionality and ability to homo/heterodimerize. However, following nanotransfer, the receptor would be in a more physiologically relevant membrane environment compared to that of the nanodisc. Thus, any artifacts of the nanodisc lipid composition that might have impacted the structure and function of the receptor in the nanodisc could have been “erased” by re-introduction to cells. Did the authors observe any evidence of lipids in their structure that might have impacted the receptor structure?*

A: We did not observe any lipid molecules in the structure of H₂R, nor we do have any other evidence of a specific lipid influence on the structure. The overlay of the H₂R complex structure

matches nicely with that of the H₁R complex (FigRev1A; RMSD of 1.41Å) within the common regions, so we do not see any evidence of artifacts. Same after comparison with previously published structures of GPCRs in nanodiscs (FigRev1B). H₂R binds histamine and forms its active complex in the nanodisc environment. After nanotransfer, we show histamine-induced internalization which requires agonist binding, so the two results are in agreement with each other. The additional interaction of the transferred H₂R with endogenously synthesized H₂R and H₁R can hardly be carried out in nanodiscs.

FigRev1: Comparison of the active structures of the H₂R with the H₁R (A) and other Class A GPCRs obtained in nanodiscs (D₂R and NTSR1) (B)

Q: 4. As the authors state, “The lipid composition of ND membranes can be of crucial importance of the efficiency of cotranslational membrane insertion as well as for the function of the inserted membrane proteins.” (Lines 74-76) Can the authors comment on why DOPG and DEPG were most efficient for H₂R insertion? Is this phenomenon related to the *E. coli* CF expression system as opposed to the requirements of the H₂R receptor? Do other membrane proteins expressed in this system exhibit a similar preference for versions of PG? If so, this could be a major limitation of this system for expression of other receptors in a functional state in similar *E. coli* cell-free expression systems.

A: As explained above, negatively charged lipids can promote the translocon-independent membrane insertion (Ridder et al., 2001). We have added a corresponding reference and rephrased the result section as mentioned above. Two different mechanisms have to be considered, the insertion efficiency and the subsequent folding efficiency. However, the supporting effect of negatively charged lipids is not a limitation of the system in view of screening lipid modulators. In order to promote membrane insertion, it is sufficient if negative charged lipids are just prevalent in the membrane. The MSP1E3D1 nanodiscs contain approx. 200 lipid molecules and a substantial fraction could be used in mixtures with PG for screening, as e.g. shown with the PG/CL or CHS mixtures in this manuscript or with other combinations in previous publications. We have added one corresponding reference into the results section (Henrich et al., 2016). It should also be considered that usually just one or few molecules of allosteric lipids are sufficient to act on a GPCR.

Q: 5. The significance of potential G_{αs} interaction with POPG lipids is unclear, since the receptor is in a non-native lipid environment. However, a similar type of interaction has been extensively documented for other receptors. As mentioned above, other more human-relevant acidic phospholipids would have been worthwhile to test.

A: Previous studies have shown that basic residues in the N-terminal αN helix of G_{αs} are important for plasma membrane localization of the G protein (Crouthamel et al., 2008) and promote G_s coupling to the β₂ adrenergic receptor by interacting with negatively charged lipid

headgroups (Strohman et al., 2019). Similar results have been obtained for other G proteins, including G α_q (Crouthamel et al., 2008; Crouthamel et al., 2010), G α_i (Yin et al., 2020; Zhang et al., 2021) and G α_{14} , and G α_{16} (Pedone and Hepler, 2007), suggesting that the positively charged residues in the N-terminal α_N helix play important roles in membrane anchoring of heterotrimeric G proteins and as a result of GPCR coupling. For the β_2 AR both negatively charged lipids, POPS and POPG, increase G $_s$ coupling to the receptor to similar extent, indicating that the negative net electrostatic charge of the phospholipids is more important for G $_s$ coupling than the chemical nature of the headgroup. However, since this could also be receptor dependent, we rephrased this part and cited G $_s$ -related references.

Lines 321-331: “Five positively charged residues, K8, K17, K24, R13, and R20, in the α_N helix of the H $_2$ R-coupled G α_s subunit are found to point towards the ND lipid bilayer to potentially interact with the polar membrane headgroups (Extended Data Fig. S10C). Notably, previous studies have shown that basic residues in the α_N helix of G α_s play important roles for the plasma membrane localization of the G protein and β_2 adrenergic receptor-mediated activation of G $_s$ in negatively charged lipid environment^{42,43}. Furthermore, similar electrostatic interactions between basic residues in the α_N helix and the lipid bilayer have been found in G $_i$ complex structures of the dopamine receptor and the neurotensin receptor 1 NTSR1⁴⁴ in NDs, demonstrating that these N-terminal polybasic regions might play an important role for membrane anchoring and receptor-mediated activation of different G protein families in agreement with previous mutagenesis and functional studies^{42,45}.”

We agree with the reviewer that it would be highly valuable to determine H $_2$ R-G $_s$ structures in different phospholipid environments. However, this is beyond the scope of this manuscript, but could be done in future studies to investigate the role of different lipids on the structure and dynamics of GPCR signaling complexes.

Q: 6. *The authors used histamine-containing H1R and H2R structures to evaluate binding mode prediction tools, settling on Autodock Vina. They then used Autodock Vina to predict binding modes for H2R-selective agonists in both receptors to understand selectivity and validated predictions by swapping receptor-specific residues predicted to be involved in agonist interaction. While this approach has provided insights into receptor agonist selectivity, more convincing insights could have been obtained by using their CF expression system to enable additional structures with these agonists. Did the authors pursue structures with other agonists, like amthamine, using their CF expression pipeline?*

A: This would certainly be an interesting approach and might be subject of further studies. However, as we present the first cell-free generated structure, our current main focus is rather to apply the process to other targets in order to demonstrate the versatility of the system and to expand its usage.

Reviewer #2 (Remarks to the Author)

The submitted article entitled “Cryo-EM structure of cell-free synthesized human histamine H2 1 receptor coupled to heterotrimeric Gs protein in lipid nanodisc environment”, by Kock1 et al., presents a tour de force approach and analysis on the structure and function of human histamine 2 receptor. The manuscript is highly focused and well written. The techniques and results are sequentially presented and easy for the reader to comprehend. Overall, the article is extremely well written, and the authors have provided detailed information, protocols, and statistical analysis of the data. All the figures reflect the authors findings and conclusions.

First, the authors show that cell-free expression is a valuable tool for screening lipids and other additives such as lipids, nanodisc and G-protein complexes. The cell-free lysates are then scaled up for production to provide material for biochemical characterization and novel structural insight into the H2 type receptors bound with ligand and G-protein. Furthermore, the authors show that this approach does not require a detergent solubilization step for the GPCRs and allowed for full length expression of the native like amino acid sequence without the need for stabilization domains/mutants. The authors were also able to provide new data on the complex at a resolution of 3.4 angstrom, which is comparable to analogous structures of GPCR complexes expressed and purified from eukaryotic cell-based systems but relied on other techniques. Regarding these other structures, the authors may have missed an opportunity to highlight the uniqueness of this approach. To biochemically validate the predicted complexes, the authors used mutagenesis for comparative studies to WT in the presence of different ligands along with G-protein activation. This approach also confirmed homo- and hetero-oligomerization as well as the impact of specific side chains on receptor function. The nanodelivery technique and data could have been exploited to provide greater information on correctly folded and functional complexes.

I believe this is an important area of research in the GPCR field and the general readers will be extremely interested in the manuscript and its findings related to novel techniques for producing membrane proteins in multiprotein complexes that are correctly folded and functional. I would recommend this paper for publication after addressing several minor specific comments (see below).

We thank the reviewer for the very positive comments about our work.

Specific Comments:

Q: 1. *It was not clear what the authors meant by “...semi-defined CF environment obtained from E. coli strain A19...? Was this referring to the lysate itself, the buffer conditions or some genetic alteration in the A19 strain?*

A: “Semi-defined” should refer to the S30 lysate composition roughly consisting out of 1.000 different proteins. Out of these 1.000 proteins, we have identified a reliable core proteome present in our lysate after applying standardized preparation procedures and covering roughly 750 different proteins (Foshag et al., 2018). However, some further 250 proteins are low abundant and their presence might vary from batch to batch, thus we keep the lysate as “semi-defined”. Buffers and any supplied low molecular weight-compounds are certainly defined.

We have rephrased the sentence in the discussion to avoid any confusion:

Lanes 396-399: “A CF lysate from *E. coli* strain A19 previously defined by proteomics analysis⁵⁴ was used in combination with supplied preformed lipid NDs, heterotrimeric G_s

protein and Nb35 to obtain a structure of the full-length H₂R-G_s complex with its endogenous agonist histamine.”

Q: 2. *The authors use the term “co-translation” multiple times in the article, but is never clear to the reader when two or more things are being co-translated?*

A: With “cotranslational” in the manuscript, we always mean the insertion of the nascent GPCR into the provided preformed nanodisc membranes. We rephrased several sentences containing this term to avoid confusion.

Q: 3. *The following papers should be considered for citation given they demonstrated the use of preformed nanodisc for generating membrane proteins such as GPCRs (Katzen, F., et al. (2008). “Insertion of membrane proteins into discoidal membranes using a cell-free protein expression approach. *Proteome Res* 7(8): 3535–42.). These same authors also demonstrated this same technique for producing full-length GPCRs, but required insertion of a T4L stabilizing domain for the GPCR of interest (Yang et al. *Cell-free synthesis of a functional G protein-coupled receptor complexed with nanometer scale bilayer discs. BMC Biotechnol* 11, 57 (2011). <https://doi.org/10.1186/1472-6750-11-57>). In my opinion, the authors should emphasize the benefit of their approach compared to previous publications that required the use of the stabilizing insertions for structural studies. How does the current manuscripts resolution compare to previously solved crystal structures?*

A: The Yang paper was already cited in our manuscript (ref. 11) in addition to two relevant papers of Katzen and coworkers (refs. 14 and 15). Additionally, we now added the proposed publication of Katzen et al. 2008. However, direct comparisons with the results of these references are difficult because different protocols, nanodisc sizes and GPCR targets are used in these studies. All available structures of GPCR/G protein complexes in nanodiscs have been solved at resolutions of > 3.4 Å. We analyzed the resolution of all available GPCR-G protein complex structures listed in the GPCRdb (www.gpcrdb.org) and found similar medians of 3.1Å for all 524 cryo-EM structures and 3.05Å for the 18 published X-ray structures (FigRev2). The resolution of our structure is positioned at the 85th percentile of both datasets (X-ray and cryo-EM). However, to obtain most of the X-ray and cryo-EM structures, non-natural detergent environments have often been used in conjunction with engineering of the receptors and/or coupled G proteins. For the H₂R/G_s complex determined in our work, no other crystal structure or cryo-EM structure is currently available for comparative analysis.

GPCR-G protein complex structures

FigRev2: Comparison of the resolution of available GPCR-G protein complex structures. The median for the resolution of structures obtained by cryoEM and X-ray crystallography is 3.1 and 3.05Å, respectively, and was calculated and plotted using GraphPad Prism.

Q: 4. Did the authors consider the opportunity to use computational structural methods for comparative studies to predicted GPCR structures? There should have been a wealth of information validating or disproving the predictive structures in databases such as AlphaFold regarding the human histamine 2 receptor.

A: Following the suggestion of reviewer 2, we performed an alignment of the model of human H₂R in complex with G α_s , as generated by the AlphaFold2 (AF2) multimer tool via the COSMIC2 platform, and the experimental structure that we obtained, using the “MatchMaker” tool of the UCSF Chimera software. The sequences for the human H₂R and Guanine nucleotide-binding protein G(s) subunit alpha were used as reported in the UniProt database. The overall superposition of backbones and side chains of the experimental and predicted complexes shows a rather high degree of congruence. Interestingly, this was not the case when the AlphaFold model of the receptor alone (AF ID: AF-P25021-F1_model_v4) was superimposed on the receptor portion of our experimental complex. In this case, transmembrane helix 6 is moved outward to a much lower degree, congruent with the absence of the G protein. The differences between the experimental complex and the AF2 model are mainly relative to the most flexible portions in the receptor portion, namely the N-termini, and the extra- and intra-cellular loop regions. The high flexibility of all these regions is a well-known feature of GPCR structures. Furthermore, small portions of ECL2, ICL3 and H8 were not resolved, due to resolution limitations discussed in the manuscript. Hence, we cannot comment on the validity of the prediction for these regions. Similarly, in the AF2 model of the G protein, the flexible region of the α -helix is shifted upwards with respect to the experimental structure, which might be due to the known limitations in the modelling software when sampling the completely active conformation of a protein, in this case of the G protein.

We modified the Fig. S3 by adding the superposition of the experimentally determined structure and the AF2 model and we added the following text into the section “Structure determination of the histamine/H₂R/Gs signaling complex” Lines 166-173:

“Additionally, we performed an alignment of our experimental structure with a computationally predicted model of the human H₂R in complex with G α_s . The model was generated by the AlphaFold2 (AF2) (Varadi et al, 2022) multimer tool via the COSMIC2² platform (Cianfrocco

et al., 2017) using the sequences for the human H₂R and guanine nucleotide-binding protein G(s) subunit alpha as reported in the UniProt database. The overall superposition of backbone atoms and side chains of the experimental and predicted complexes shows a rather high degree of congruence, with a C α -RMSD of the receptor portion of 1.046 Å. (Extended data Fig. S3).”

Reviewer #3 (Remarks to the Author)

The manuscript by Köck Z et al. describes the cryo-electron microscopy (cryoEM) three-dimensional (3D) structure of the active human histamine H2 receptor (H2R) in complex with its natural ligand histamine, the heterotrimeric Gs protein, and the stabilizing nanobody Nb35. It should be emphasized that this is the first structure of a G protein-coupled receptor (GPCR)/G protein signaling complex in which the GPCR is produced using a cell-free system (*E. coli* lysate) supplemented with purified Gs protein and Nb35. The quantities of H2 receptor are compatible with structural analysis. The study enabled single-particle analysis of the complex at a global resolution of 3.4 Å allowing to provide a structural basis for the activation and ligand recognition of this histamine H2 receptor which plays a central role in diverse pathophysiological situations. Again, the H2R is cotranslationally inserted into preformed nanodiscs (ND), and it is demonstrated it can be directly transferred from these particles to living cell (HEK293 cells) plasma membranes. The transfer is efficient since it is shown the receptor is internalized upon ligand binding and is able to form heterodimers with transfected histamine H1 receptors. Finally, the determination of the H2R/Gs/Nb35 complex structure allowed the authors to understand what makes the specificity of several H2-selective ligands to H2R, relative to other histamine receptors.

This work is a nice combination of cell-free GPCR production, cryoEM structure analysis, nanotransfer of GPCR from ND particles to living cells, mutagenesis, signaling assays, and ligand docking approaches. This strategy should become in the future more broadly applicable to other GPCRs or other membrane protein families. Regarding the H2R, it may help in the rational design and development of novel H2R-selective ligands for treating gastric acid secretion, cardiac activity and vasodilatation.

The expertise of the authors in the scientific field of GPCRs and molecular pharmacology is internationally well renowned, the methods developed are state-of-the-art and the results presented in the article are really interesting for GPCR's research community. However, I have some concerns and also some minor comments before the manuscript would be accepted for publication in Nature Communications.

We thank the reviewer for putting our work into context and for the positive evaluation of its importance.

1/ major comments:

Q: a) My first criticism is focused on the mutagenesis study and more particularly on assessment of receptor expression levels to analyze BRET-based G protein dissociation (which means activation in that case). It is not sufficiently described, neither in methods nor in figure legends (Fig 4b and c, Fig 6c and d, Fig S7). I understood the expression level of some mutants is very low and that of wild-type (WT) has to be adjusted to compare ligand-induced G protein activation in a rigorous way. Results are shown as dose-response curves of Gs or Gq activation depending on ligand concentration and are measured as deltaBRET. This is OK. But, for instance, in the panels b and c of figure 4, 50% or 10% of WT (H1R or H2R) are used in order to compare correctly receptor expression levels between wild-type and mutants. What does mean 50%, 10%? What is the 100%? Unfortunately, Fig S7 does not help to better understand these values. What are the quantities of plasmid transfected for each WT and mutant receptor to manage comparable levels of expression? This should be indicated in Methods. Moreover, in Figure 4, the H2R mutant T190N is analyzed whereas in Fig 6 it is explained that it is not included because of its very low expression level. Thus, is this mutant correctly expressed or not? This is not clear.

A: We thank the reviewer for pointing this out. For the BRET experiments, we used a total amount of 1 μg of DNA per ml of resuspended cells to be transfected, which corresponds to 100%. This amount includes 500 ng of G protein sensor DNA (50%) and up to 500 ng of receptor DNA (50%). To compensate for the different expression levels of the receptor mutants shown in Fig. S7, we performed a titration of WT receptor DNA (100-500 ng corresponding to 10% - 50%) to achieve the same expression level of the mutants and added empty pcDNA to adjust the total DNA amount to 1 μg . We have now added a corresponding explanation to the method section and to the figure legends of Figs. 4, 6 and S7. Furthermore, we replaced the percentage with the amount of DNA/mL cell culture in Figs. 4, 6 and S7.

For our first pharmacological characterization of the H₂R mutants, we wanted to include all mutants generated regardless of the measured surface expression levels because the ELISA-based expression assay is much less sensitive in detecting low receptor levels that are able to promote signaling. Hence, we also included the H₂R mutant T190N in the BRET assay shown in Fig. 4, to exclude that low but in the ELISA undetectable levels of this mutant are present and able to promote signaling (as seen for D186T). As this mutant showed no histamine-dependent G protein dissociation and no to very low surface expression, it was not included in the subsequent BRET assays to test the H₂R-selective ligand. In contrast, the D186T mutant was included in all assays, although it also showed very low surface expression. However, other than T190N, its activity was comparable to the 10% H₂R (WT) data with similar receptor levels in the plasma membrane. As mentioned above, this suggests that the ELISA assay shown in Fig. S7 is not very sensitive at detecting very low receptor expression levels. In contrast, these low levels of receptor surface expression are sufficient to show a robust signal in the BRET-based G protein activation assay as shown for the D186T mutant. We changed the sentence in the main text to make this clearer.

Lanes 292-293: “Notably, the H₂R mutant T190N^{5,46} was excluded from these experiments because of its lack of expression and inactivity (Extended Data Fig. S7).”

Q: b) My second criticism is about nanotransfer of H₂R into living cells from the nanodiscs to the plasma membrane of these cells. The strategy is strikingly interesting and promising in the case of GPCRs or other membrane proteins which are difficult to endogenously express through transfection of DNA. In the present study, efficient transfer of the nanodisc-embedded receptor into plasma membranes of HEK293 cells is demonstrated through ligand-induced internalization (measured with fluorescence of monomeric neogreen protein fused at the C-terminus of H₂R). Colocalization with transfected H1R is also shown, as well as heterodimerization with this related histamine receptor.

To my opinion, at least two assays should be added to complete the study. To show that the H₂R-transferred is totally functional and possess equivalent properties to H₂R synthesized in mammalian cells, a comparison with a transfected H₂R may be done. In other words, determination of the K_d of histamine for the H₂R with cells expressing H₂R after transfection or expressing H₂R after nanodisc-membrane transfer could be performed using ligand binding assays (radioactivity or FRET approaches).

In addition, accumulation of cAMP upon histamine binding (dose-response curves) to HEK293 cells transfected with H₂R plasmid or transferred with H₂R/ND particles should be compared to check whether the EC₅₀ values are in the same range.

A: We agree with the reviewer that it will be highly valuable to understand the functional properties of nanotransferred H₂R in HEK cells. However, it needs to be considered that the efficiency of nanotransfer is much lower if compared with transfection of a plasmid-encoded H₂R (approx. 10%). A further challenge is that only 50% of the transferred H₂R is inserted with a correct orientation as the transfer is random. The technique is still very new and by far not as established as transfection. Quantitative assays as being common for transfected cells still need

to be optimized by increasing sensitivity or GPCR transfer rates. We performed BRET-based cAMP assays with H₂R-transferred HEK293A cells using the CAMYEL (cAMP sensor using YFP-Epac-Rluc) sensor (FigRev3). Cells that were transferred with H₂R nanodiscs showed a decreased BRET signal relative to cells treated with empty nanodiscs (statistically significant difference confirmed using extra-sum-of-squares F-test comparing the plateaus of both dose-response curves), indicating that the nano-transferred receptor is functional and induces a Gs-dependent increase in cAMP. However, due to the low transfer efficiency, the signal intensity was significantly smaller than for the cotransfected H₂R. It is also noteworthy that the EC₅₀ values determined in these functional assays are highly dependent on the number of activated receptors embedded in the cell membrane (more receptors at the membrane left-shift the dose-response curves). Thus, differences in EC₅₀ between cotransfected H₂R and nanotransferred H₂R do not necessarily represent impaired function of nanotransferred H₂R, but are likely to be a consequence of low transfer efficiency.

FigRev3: BRET-based cAMP assay of HEK293A cells transferred with empty nanodiscs (NDs) or H₂R NDs in comparison to H₂R-transfected cells.

Unfortunately, we were not able to perform radioassays due to the limited availability of labeled histamine and the reported low histamine binding affinity for H₂R.

Q: *c) The third major criticism deals with ligand docking calculations. The comparison of the reported cryoEM H₂R (in complex with histamine) structure with that of H₁R (also in complex with histamine) already available gives very important information to understand differences in histamine docking within the two related receptor orthosteric pockets, and more importantly to determine what makes the selectivity of H₂-selective compounds for H₂R. Indeed, the authors demonstrated that divergent residues at positions 5.42 and 5.46 (Ballesteros-Weinstein nomenclature) in TM5 were identified to participate in the agonist binding selectivity of the H₂R.*

The docking procedures are more questionable, in particular the SEED predictions. Indeed, I do not understand the different pictures of Fig S9a. First the legend of this figure is too short and does not help in complete interpretation of the data. If the legend of panel a is equivalent to that of panel b, this would mean that predicted binding poses are colored in sand whether

those which are experimentally resolved are in orange. However, there is only one histamine binding pose described in the 3D structure of the histamine-H2R/Gs/Nb35 complex, not 6, even not 2 as indicated in the legend (top-2-scored binding modes are reproducing the experimentally resolved orientations). Do the different pictures in Fig 9a correspond to different iterations of the docking molecular simulations? Or do they correspond to different clusters of binding poses?

I would remove the Figure S9 panel a, which is not necessary, taking account all other data presented in the manuscript. I would keep only the AutoVina docking predictions. In addition, Table S2 and S3 should have captions to explain the different steps in the docking procedures and calculation of binding energy constants.

A: We apologize for the brevity of the caption of Fig. S9a, which was indeed not transmitting the intended message. In the revised version, we have rephrased the figure caption as follows (Lines 1191-1199): “The multiple-copy docking software SEED was used to predict possible alternative binding modes (orange carbons) for histamine in the H2R in order to assess the likelihood of an alternative orientation compatible with the experimental density. In the orange-framed boxes, the SEED energy score values are reported. The SEED energy score takes into account desolvation and can thus be assumed to yield accurate predictions. Shown are the representatives of the six clusters calculated by SEED in order of descending score. The top-2-scored binding modes are reproducing the experimentally resolved orientation of histamine (yellow carbons), lending additional validity to the experimental orientation.”

Concerning Tables S2 and S3, we added a redirection to the Methods section where we added information about the main differences between the utilized docking softwares. References to all sources, including the software development articles, are reported in the same section. The following text has been added to Methods (Lines 828-845): “The following softwares were used for docking calculations: Autodock Vina (Trott et al., 2011), DOCK3.7 (Coleman et al., 2013), the OpenEye programs FRED and HYBRID (McGann et al., 2012), and SEED (Solvation Energy for Exhaustive Docking (Majeux et al., 1999)). The differences among these softwares lie in both the search algorithms used to sample the different orientations of the ligands in the binding pockets and the energy functions with which these orientations are scored (scoring functions). Autodock Vina applies the Lamarckian genetic algorithm to generate and optimize the possible orientations, thus accounting for ligand flexibility. In contrast, DOCK uses a shape matching method to sample the different ligand conformations, thus treating them as individual rigid bodies. OpenEye programs FRED and HYBRID also consider each ligand’s flexibility by pre-generating different conformations using the program OMEGA. HYBRID additionally uses information from the experimental pose of the ligand, when available. Scoring functions are different for each of the considered softwares: A knowledge-based scoring function is applied by Autodock Vina for the ranking of poses, while force field-based and empirical-based functions are applied by DOCK and OpenEye softwares, respectively. The SEED software, differently from all the previously mentioned ones, is used for docking of fragments, and similar to DOCK, its search algorithm is based on exhaustive matching of conformers to the binding region. It also uses a force field-based scoring function, which gives particular emphasis to protein and fragment desolvation upon binding.”

2/ minor comments: there are many.

Q: *a) Is the monomeric neongreen (NG) fluorescent protein still present in the H2R construct for cryoEM analysis of the particles?*

A: No, the H₂R-mNG derivatives were only used for monitoring the nanotransfer. We now better indicated that in the results section and in the caption of Fig. S1. We rephrased the result section as follows (Lanes 99-109):” All ND types were supplied at final concentrations of 60

μM and reaction concentrations of solubilized H₂R-mNG between 5 μM and 35 μM were obtained (Extended Data Fig. S1A). The negatively charged lipids DOPG or DEPG were identified as being most efficient for H₂R-mNG insertion (Extended Data Fig. S1A). Besides quantity, the quality of the synthesized GPCR is of crucial importance and size-exclusion chromatography (SEC) can be used to separate functionally folded GPCR/ND particles from soluble but aggregated fractions¹³. SEC analysis was performed with CF synthesized H₂R without the mNG moiety and purified H₂R/ND complexes showed the best sample quality with ND membranes composed of DOPG, DEPG and DMPG (Extended Data Fig. S1B). SEC profiling was then used to monitor additional effects of various supplied ligands on the quality of cotranslationally synthesized H₂R/ND (DOPG) complexes (Extended Data Fig. S1C).”

Q: *Palmitoylation of cysteine 305 is not performed in the cell-free production system, leading probably to a disordered receptor C-terminus. Would the presence of the mNG fluorescent module participate in this increased flexibility of the protein and to conformational heterogeneity? This should be discussed.*

A: The mNG moiety was not present in the structural experiments. As mentioned above, we have now indicated that in the results section and in the caption of Fig. S1. We also agree with the reviewer that it might have caused problems. At this point we want to emphasize that cell-free expression in *E. coli* lysates does not require the attachment of any large fusion partners (e.g. MBP, NusA, Trx) as it is frequently the case upon GPCR expression in *E. coli* cells.

Q: *b) Page 5, line 211: please write 7UL3 instead of 7IL3.*

A: Thanks for pointing this out, it is now corrected.

Q: *Is the affinity of histamine equivalent for H₁R and H₂R? it should be indicated in the paragraph “comparison of histamine binding between the H₂R and H₁R”. References should be added.*

A: The affinity of histamine to H₂R (pK_i 6.6) is approx. 10fold higher if compared with the affinity to H₁R (pK_i 5.6). We added this statement and corresponding references in the recommended paragraph. Lanes 185-186: “The affinity of histamine to H₂R (pK_i 6.6) is approx. 10fold higher if compared with the affinity to H₁R (pK_i 5.6)^{33,34}.”

Q: *d) Page 7, second paragraph (lines 311-323): the Extended Data Figure S10D should be referred somewhere in this paragraph.*

A: We have now included this reference in the text.

Q: *e) Page 8, line 356: please replace Fig. 7C with Fig8C.*

A: We have now fixed this.

Q: *f) Fig S1: in panel a, insertion of H₂R into nanodiscs with different lipids is shown and its concentration measured using fluorescence of mNG. Does this mean that in some conditions, half of the preformed discs can insert the synthesized H₂R (30 μM of receptor versus 60 μM of ND)?*

A: What we measure in this initial assay is the concentration of folded mNG attached to the synthesized GPCR in the supernatant of the reaction after removal of precipitates by centrifugation. As each target/DNA template results into an individual expression rate, the final concentration of the supplied NDs is titrated until maximal fluorescence is obtained in the CF reaction supernatant. For H₂R, that was 60 μM NDs (DOPG). We included a corresponding statement and better explanation in the first paragraph of the results section (Lanes 102-112). The finally determined H₂R concentration was estimated to 30 μM , meaning that maximal half of the supplied NDs are associated with H₂R-mNG if we assume a 1:1 association. However,

as supported by the SEC profiles in panel b, a significant fraction of the synthesized H₂R-mNG/ND particles is present in some form of soluble aggregates and may contain even multiple associated H₂R-mNG. The initial contact between nascent GPCR and an empty ND is presumably a crucial limiting parameter to initiate membrane insertion and functional folding. Over-titration of the GPCR with the supplied empty NDs is thus necessary to obtain sufficient sample quality.

Q: *What means c in c(H₂R-mNG) [uM]?*

A: C in c(H₂R-mNG) [μM] means concentration of the H₂R-mNG in μM. We added a comment in the figure caption.

Q: *In panel b, it would be interesting to show a SEC profile of empty nanodiscs for comparison, at least that with DOPG, since this lipid has been chosen for subsequent studies (DMPG is also interesting since the fraction of proposed folded H₂R seems to be well isolated).*

A: As suggested by the reviewer, we have added the SEC profile of empty DOPG NDs to panel b. This SEC profile is indeed representative for empty NDs formed with MSP1E3D1 in general, since it usually does not depend on the lipid composition of the NDs.

Q: *Moreover, relative absorbance is indicated in the y axis: is the value different for each H₂R/ND complex? It is not indicated. This would be useful for comparing the yields of H₂R/ND purification. I have the same remark for panel c.*

A: In case of panel b: The purification yield is correlated to the overall yields of the various H₂R/ND complexes shown in panel a. For example, DMPG shows a slightly more pronounced peak at an elution volume of 1.5mL than DOPG. However, the expression yield of DOPG is approx. 50% higher than that of DMPG (panel a), thus resulting in a higher yield of folded H₂R in DOPG NDs.

In case of panel c: Addition of ligands to the CF reaction does not change the CF expression yields. Therefore, the absorbance values are similar and the peaks at an elution volume of 1.5mL can be directly compared. We added a statement about the expression rates for panel b and c into the figure caption.

Q: *At the end of the legend, void volume is abbreviated by V. In the corresponding figure, V is not mentioned. I guess V is indicated with the arrow.*

A: The assumption is correct and the description was changed in the legend.

Q: *g) Fig S5: the density of TM7 is shown on the left. Corresponding numbering indicates amino-acids from 267 to 323. This would mean a density for TM7-helix8 and part of the C-terminus. This is not possible. Please correct.*

A: We thank the reviewer for the careful evaluation and appreciate the feedback regarding Figure S5. We have implemented the necessary corrections and the density shown on the left now accurately represents TM7 residues from 267-304.

Q: *h) Fig S6: in the sequence alignment, asterisks indicate cysteines involved in disulfide bridges. Cysteine 246 of H₂R is not involved in a disulfide bridge with cysteine 174. Please put an asterisk for cysteine 91 instead of cysteine 246.*

A: We apologize for that mistake. We have now replaced Fig. S6 with a modified version indicating the correct cysteines as suggested.

Q: *i) Fig S9: In the title of this figure, please remove “and BRET assays”, since the figure only deals with histamine docking poses.*

A: This has been corrected.

References

Cianfrocco, M. A., Wong-Barnum, M., Youn, C., Wagner, R. & Leschziner, A. COSMIC2: A science gateway for cryo-electron microscopy structure determination. *Proceedings of the practice and experience in advanced research computing 2017 on sustainability, success and impact.* 22, 1-5 (2017).

Coleman, R. G., Carchia, M., Sterling, T., Irwin, J. J. & Shoichet, B. K. Ligand pose and orientational sampling in molecular docking. *PLoS ONE* 8, e75992 (2013).

Crouthamel, M. et al. N-terminal polybasic motifs are required for plasma membrane localization of Gas and Gαq. *Cell. Signal.* 20, 1900-1910 (2008).

Crouthamel, M. et al. An N-terminal polybasic motif of Gαq is required for signaling and influences membrane nanodomain distribution. *Mol. Pharmacol.* 78, 767-777 (2010).

Foshag, D. et al. The E. coli S30 lysate proteome: Prototype for cell-free synthetic biology. *New Biotechnol.* 40, 245-260 (2018).

Fukushima, Y. et al. Structural and functional analysis of the canine histamine H2 receptor by site-directed mutagenesis: N-glycosylation is not vital for its action. *Biochem. J.* 310, 553-558 (1995).

Fukushima, Y. et al. Palmitoylation of the canine histamine H2 receptor occurs at Cys305 and is important for cell surface targeting. *Biochim. Biophys. Acta (BBA)-Molecular Cell Research* 1539, 181-191 (2001).

Henrich, E. et al. Lipid requirements for the enzymatic activity of MraY translocases and in vitro reconstitution of the lipid II synthesis pathway. *J. Biol. Chem.* 291, 2535-2546 (2016).

Inagaki, S. et al. Modulation of the interaction between neurotensin receptor NTS1 and Gq protein by lipid. *J. Mol. Biol.* 417, 95-111 (2012).

Köck, Z. et al. Biochemical characterization of cell-free synthesized human β1 adrenergic receptor cotranslationally inserted into nanodiscs. *J. Mol. Biol.* 434, 167687 (2022).

Kofuku, Y. et al. Functional dynamics of deuterated β2-adrenergic receptor in lipid bilayers revealed by NMR spectroscopy. *Angew Chem.* 126, 13594-13597 (2014).

Majeux, N., Scarsi, M., Apostolakis, J., Ehrhardt, C., & Caflisch, A. Exhaustive docking of molecular fragments on protein binding sites with electrostatic solvation. *Proteins* 37, 88-105 (1999).

- McGann, M. FRED and HYBRID docking performance on standardized datasets. *J. Comput-Aided Mol. Des.* 26, 897-906 (2012).
- Merk, H. et al. Biosynthesis of membrane dependent proteins in insect cell lysates: identification of limiting parameters for folding and processing. *Biol. Chem.* 396, 1097-1107 (2015).
- Mio, K. & Sato, C. Lipid environment of membrane proteins in cryo-EM based structural analysis. *Biophys. Rev.* 10, 307-316 (2018).
- Pedone, K. H. & Hepler, J. R. The importance of N-terminal polycysteine and polybasic sequences for G14 α and G16 α palmitoylation, plasma membrane localization, and signaling function. *J. Biol. Chem.* 282, 25199-25212 (2007).
- Ridder, A. N. J. A., Kuhn, A., Killian, J. A., & de Kruijff, B. Anionic lipids stimulate Sec-independent insertion of a membrane protein lacking charged amino acid side chains. *EMBO Rep.* 21, 403-408 (2001).
- Staus, D. P. et al. Detergent-and phospholipid-based reconstitution systems have differential effects on constitutive activity of G-protein-coupled receptors. *J. Biol. Chem.* 294, 13218-13223 (2019).
- Strohman, M. J. et al. Local membrane charge regulates β 2 adrenergic receptor coupling to Gi3. *Nat. Commun.* 10, 2234 (2019).
- Thakur, N. et al. Anionic phospholipids control mechanisms of GPCR-G protein recognition. *Nat. Commun.* 14, 794 (2023).
- Trott, O. & Olson, A. J. AutoDock Vina: Improving the speed and accuracy of docking with a new scoring function, efficient optimization and multithreading. *J. Comput. Chem.* 31, 455-461 (2011).
- Umbach, S. et al. Transfer mechanism of cell-free synthesized membrane proteins into mammalian cells. *Front. Bioeng. Biotechnol.* 10, 906295 (2022).
- Unsay, J. D., Cosentino, K., Subburaj, Y. & García-Sáez, A. J. Cardiolipin effects on membrane structure and dynamics. *Langmuir* 29, 15878-15887 (2013).
- Varadi, M. et al. AlphaFold Protein Structure Database: Massively expanding the structural coverage of protein-sequence space with high-accuracy models. *Nucleic Acids Res.* 50, D439-D444 (2022).
- Whorton, M. R., et al. A monomeric G protein-coupled receptor isolated in a high-density lipoprotein particle efficiently activates its G protein. *Proc. Natl Acad. Sci.* 104, 7682-7687 (2007).
- Yin, J. et al. Structure of a D₂ dopamine receptor-G protein complex in a lipid membrane. *Nature* 584, 125-129 (2020).
- Zhang, M. et al. Cryo-EM structure of an activated GPCR-G protein complex in lipid nanodiscs. *Nat. Struct. Mol. Biol.* 28, 258-267 (2021).

REVIEWERS' COMMENTS

Reviewer #1 (Remarks to the Author):

The authors have responded adequately to my questions and comments, although some of my concerns remain surrounding the physiological relevance of their lipid choices. It also would have been nice to see this novel cell-free expression approach applied to additional targets to show its robustness beyond histamine receptors. Overall, however, I support the publication of this manuscript in Nature Communications.

Reviewer #2 (Remarks to the Author):

The submitted article entitled "Cryo-EM structure of cell-free synthesized human histamine H2 1 receptor coupled to heterotrimeric Gs protein in lipid nanodisc environment", by Kock1 et al., presents a tour de force approach and analysis on the structure and function of human histamine 2 receptor. The article is extremely well written, and the authors have provided detailed information, protocols, and statistical analysis of the data. All the figures reflect the authors findings and conclusions. Overall, the authors have provided a detailed response to the reviewers that addresses any concerns I had with the manuscript. I believe this is an important area of research in the GPCR field and the general readers of Nature Communications will be extremely interested in the manuscript and its novel findings. I would highly recommend this paper for publication in its current form.

Reviewer #3 (Remarks to the Author):

The revised version of the manuscript by Köck Z et al. describes the cryo-electron microscopy (cryoEM) three-dimensional (3D) structure of the active human histamine H2 receptor (H2R) in complex with its natural ligand histamine, the heterotrimeric Gs protein, and the stabilizing nanobody Nb35. This is the first structure of a G protein-coupled receptor (GPCR)/G protein signaling complex in which the GPCR is produced using a cell-free system (E. coli lysate) supplemented with purified Gs protein and Nb35. The quantities of H2 receptor are compatible with structural analysis. The study enabled single-particle analysis of the complex at a global resolution of 3.4 Å allowing to provide a structural basis for the activation and ligand recognition of this histamine H2 receptor which plays a central role in diverse pathophysiological situations. Again, the H2R is cotranslationally inserted into preformed nanodiscs (ND), and it is demonstrated it can be directly transferred from these particles to living cell (HEK293 cells) plasma membranes. Although the transfer is not highly efficient, the receptor is functional, is

internalized upon ligand binding and is able to form heterodimers with transfected histamine H1 receptors. Finally, the determination of the H2R/Gs/Nb35 complex structure allowed the authors to understand what makes the specificity of several H2-selective ligands to H2R, relative to other histamine receptors.

All the criticisms and comments I raised previously were adequately addressed in this new version. Also, I acknowledge that the remarks from the two other reviewers were correctly taken into account.

Regarding my second major criticism, i.e. the efficiency of nanotransfer of nanodisc-embedded H2R receptor into living cells, please indicate in the Method's corresponding paragraph that this approach is much less efficient than transfection with a plasmid-encoded H2R (approximately 10%). Moreover, also add that only 50% of the transferred H2R is correctly oriented in the cell membranes and that this technique is new and still has to be optimized.

I consider the manuscript is now acceptable for publication in Nature Communications.

Response to reviewers' comments

REVIEWERS' COMMENTS

Reviewer #1 (Remarks to the Author):

The authors have responded adequately to my questions and comments, although some of my concerns remain surrounding the physiological relevance of their lipid choices. It also would have been nice to see this novel cell-free expression approach applied to additional targets to show its robustness beyond histamine receptors. Overall, however, I support the publication of this manuscript in Nature Communications.

A: We thank the reviewer for the positive answer and the recommendation to publish.

Reviewer #2 (Remarks to the Author):

The submitted article entitled "Cryo-EM structure of cell-free synthesized human histamine H2 1 receptor coupled to heterotrimeric Gs protein in lipid nanodisc environment", by Kock1 et al., presents a tour de force approach and analysis on the structure and function of human histamine 2 receptor. The article is extremely well written, and the authors have provided detailed information, protocols, and statistical analysis of the data. All the figures reflect the authors findings and conclusions. Overall, the authors have provided a detailed response to the reviewers that addresses any concerns I had with the manuscript. I believe this is an important area of research in the GPCR field and the general readers of Nature Communications will be extremely interested in the manuscript and its novel findings. I would highly recommend this paper for publication in its current form.

A: We thank the reviewer for the positive evaluation of our manuscript and for the recommendation to publish.

Reviewer #3 (Remarks to the Author):

The revised version of the manuscript by Köck Z et al. describes the cryo-electron microscopy (cryoEM) three-dimensional (3D) structure of the active human histamine H2 receptor (H2R) in complex with its natural ligand histamine, the heterotrimeric Gs protein, and the stabilizing nanobody Nb35. This is the first structure of a G protein-coupled receptor (GPCR)/G protein signaling complex in which the GPCR is produced using a cell-free system (*E. coli* lysate) supplemented with purified Gs protein and Nb35. The quantities of H2 receptor are compatible with structural analysis. The study enabled single-particle analysis of the complex at a global resolution of 3.4 Å allowing to provide a structural basis for the activation and ligand recognition of this histamine H2 receptor which plays a central role in diverse pathophysiological situations. Again, the H2R is cotranslationally inserted into preformed nanodiscs (ND), and it is demonstrated it can be directly transferred from these particles to living cell (HEK293 cells) plasma membranes. Although the transfer is not highly efficient, the receptor is functional, is internalized upon ligand binding and is able to form heterodimers with transfected histamine H1 receptors. Finally, the determination of the H2R/Gs/Nb35 complex structure allowed the authors to understand what makes the specificity of several H2-selective ligands to H2R, relative to other histamine receptors.

All the criticisms and comments I raised previously were adequately addressed in this new version. Also, I acknowledge that the remarks from the two other reviewers were correctly taken into account.

Regarding my second major criticism, i.e. the efficiency of nanotransfer of nanodisc-embedded H2R receptor into living cells, please indicate in the Method's corresponding paragraph that this approach is much less efficient than transfection with a plasmid-encoded H2R (approximately 10%). Moreover, also add that only 50% of the transferred H2R is correctly oriented in the cell membranes and that this technique is new and still has to be optimized.

I consider the manuscript is now acceptable for publication in Nature Communications.

A: We thank the reviewer for the positive evaluation and recommendation to publish. As suggested, we added the following statement into the methods section:

“The nanotransfer is a recently developed approach and several persistent limitations must still be considered^{20,21}. Nanotransfer is currently much less efficient if compared with conventional transfection, reaching approx. only 10 % of a corresponding transfection efficiency. In addition, the transfer is not directed and, depending on the individual topology of the target, a significant fraction or even the majority of the transferred protein will become inserted in wrong orientation²⁰.”